



# Spatializing Net Ecosystem Exchange in the Brazilian Amazon biome using the JULES model and vegetation properties

Amauri C. Prudente Junior[1], Luiz A.T. Machado[1,5], Felipe S. Silva[1], Tercio Ambrizzi [2], Paulo Artaxo[1],

Santiago Botia[3], Luan P. Cordeiro[1], Cleo Q. Dias Junior[4], Edmilson Freitas[2], Demerval S. Moreira[6],

Christopher Pöhlker[5], Ivan M. C. Toro[1], Xiyan Xu[7], and Luciana V. Rizzo[1]

[1] Physics Institute, University of São Paulo, São Paulo, Brazil

[2] Institute of Astronomy, Geophysics and Atmosphere Science, University of São Paulo, São Paulo, Brazil

[3] Max Planck Institute for Biogeochemistry, Jena, Germany

[4] Federal Institute of Education, Science, and Technology of Pará, Belém, Brazil

[5] Max Planck Institute for Chemistry, Mainz, Germany

[6] Department of Physics and Meteorology, State University of São Paulo, Bauru, Brazil

[7] Chinese Academy of Sciences, Beijing, China

*Corresponding to*: Amauri Cassio Prudente Junior (email: amauri.cassio@usp.br)

**Abstract.** The large extension and diversity of the Brazilian Amazon biome hampers the assessment of regional-scale carbon budget based solely on local observations. Considering the shortage of observations, this study aims to examine the carbon fluxes throughout the Brazilian Amazon biome using the process-based model (JULES, Joint UK land environment simulator). A sensitivity analysis detected five critical model parameters for the Amazon tropical broadleaf evergreen forest, optimized using carbon flux and meteorological data from four forest sites. The simulations with new parametrization were compared to JULES default parameter values and with simulations of the Vegetation Photosynthesis and Respiration Model (VPRM). Net ecosystem exchange (NEE) and gross primary production (GPP) estimates were improved at all sites, reaching a Root Mean Squared Error (RMSE) about 30% lower in comparison to the default version. The optimized parameter values varied among the four sites, indicating that a single parameterization for the whole Amazonia may not be adequate. JULES model parameters were extrapolated for the Brazilian Amazonia, based on canopy height and leaf area index gridded data. Applying JULES with spatial dependent parameterization for the year of 2021 resulted in a carbon sink of -1.34 Pg C year$^{-1}$. Regional differences were observed in the carbon fluxes, with a carbon source of $0.75 \times 10^{-12}$ Pg C m$^{-2}$ year$^{-1}$ in the southwest and north, likely explained by increased ecosystem respiration in older and taller forests.



## 1. Introduction

The Amazon forest is one of the largest carbon reservoirs in the world, being relevant to the global environment, biodiversity, and climate regulation (Brienem et al., 2015). Amazon forests are responsible for 16% of the gross primary production in terrestrial ecosystems, storing approximately 90 Pg C in above- and below-ground vegetation biomass (Saatchi et al., 2011; Malhi et al., 2021). The region's critical role in the global carbon budget is at risk, as the carbon dynamics are being significantly impacted by climate change, including rising air temperatures and increased hydric stress (Liu et al., 2017; Gatti et al., 2021). These effects can lead to a decrease in the leaf area index (LAI) and an increase in plant respiration (Meir et al., 2008) and hence influence the sign of the net carbon exchange, shifting areas from sink to a source of carbon.

Accurate estimates of carbon fluxes are crucial for understanding how the Amazon will evolve under the impacts of climate change. The diverse vegetation of the Amazon biome and the strategies used to estimate carbon fluxes across different sites are essential for identifying the region's different behaviors (Restrepo-Coupe et al., 2013).The traditional method of carbon flux measurement is the Eddy Covariance (Baldochi, 2003), which quantifies the Net Ecosystem Exchange (NEE), by measuring the turbulent $CO_2$ exchange and correcting for canopy storage. NEE represents the difference between the gross primary production (GPP) of the vegetation and emissions from the ecosystem respiration (Reco) (Hayek et al., 2018). However, eddy covariance measurements are insufficient to represent the vast diversity of ecosystems and vegetation in the Brazilian Amazon biome (Aguirre-Gutierrez et al., 2025). This limitation arises due to logistical challenges, the substantial investment required for installation and equipment, and the need for highly skilled labor to ensure proper maintenance (Andreae et al., 2015). Considering the limitation of expanding flux towers throughout the Amazon biome, process-based and data-driven models have been applied in different studies to estimate NEE in different parts of Amazon, such as the Vegetation Photosynthesis and Respiration Model (VPRM) (Botia et al., 2022 and 2024), FluxCom (Nelson et al., 2024; Chen et al., 2024) and the Organizing Carbon and Hydrology in Dynamic Ecosystems (ORCHIDEE) (Verbeeck et al., 2011).

One of the comprehensive land surface models used to simulate the biophysical process is the Joint UK Land Environment Simulator (JULES; Best et al., 2011). JULES is a community land surface model used both as a standalone system and as the land surface component of the Met Office Unified Model. It is considered the state-of-the-art for large-scale simulations (Moreira et al., 2013). JULES has a tiled model of sub-grid heterogeneity being able to reproduce energy, water, carbon, and momentum fluxes (Best et al., 2011, Clark et al., 2011). The model includes up tonine land surface types containing five plant functional types (PFTs) and four non-vegetation types. Currently, JULES is used to simulate carbon fluxes in different biomes types, as applied for agriculture (Osborne et al., 2015; Williams et al., 2017) and in tropical forests (Moreira et al., 2013; Restrepo-Coupe et al., 2017; Caen et al., 2022).

Although JULES has been widely used in various studies to estimate carbon fluxes in tropical regions, a lack of specific parameterizations remains a challenge to simulate plant-soil-atmosphere interactions. Harper et al. (2016) introduced a PFT specific to tropical forests, but this parameterization has not been thoroughly tested or validated across different regions of the Amazon. Additionally, the most sensitive parameters in this region have not been deeply evaluated. In general, studies to estimate NEE using process-based models have not accounted for the large differences of vegetation characteristics across



this territory (Ometto et al., 2023). Based on these aspects, this study aims to characterize the seasonal and spatial variability of carbon fluxes between the biosphere and atmosphere in the Brazilian Amazon biome. Here, we present an improvement of the JULES parameterization specifically for the Brazilian Amazon, performing a sensitivity analysis of the model parameters having as reference to Eddy-covariance towers in different regions of the Brazilian Amazon biome. Model parameters were

spatialized using two ancillary datasets— canopy height and LAI—to access regional differences in NEE in the Brazilian Amazon biome.

## 2.Material and Methods

The current study combined observational datasets and modeling. Section 2.1 describes the study area and the tower flux sites in the Brazilian Amazon Basin. Section 2.2 describes the meteorological and edaphological datasets used as input for the JULES run and eddy-covariance dataset used to validate the model optimization.  Section 2.3 describes the gridded data used for simulations in the Brazilian Amazon biome. Section 2.4 describes the JULES model, sensitivity analysis and calibration procedures and the description of the remote sensing data and the regression method used to extrapolate the JULES

model parameters across the Amazon Basin. Section 2.5 describes the VPRM model that was used to compare with JULES model.

### 2.1. Study area

The study area corresponds to the Brazilian Amazon biome, covering 4,212,472 km$^2$. We compiled data from five eddy covariance towers to represent carbon flux and evaluate JULES simulations (Figure 1). From east to west and north to south, these sites are: The Amazon Tall Tower Observatory (ATTO), the Tapajos National Forest (K67, K83), the Reserva Jarú (RJA) and the Reserva Cuieiras near Manaus (K34). The equatorial forest was represented by 4 towers (ATTO, K34, K67, K83) and RJA represented the southern Amazonia (Restrepo-Coupe et al., 2021). The K34 tower is located 60 km north

of the city of Manaus (Araujo et al., 2002; Restrepo-Coupe et al., 2013) (Table 1).  The Santarem moist tropical forest (sites K67 and K83) is located at the confluence of the Amazon and Tapajós rivers, in the northeast Brazilian Amazon. The ATTO tower is the most recent tower built in the Amazon region, based 150 km northeast of the city of Manaus (Andreae et al., 2015). In the southern Amazon region, the RJA tower is located in a forest reserve in the state of Rondônia, characterized as a tropical wet and dry forest (Peel et al., 2007). Some of these flux towers are still operational, while others were discontinued. As such,

observations from each tower are available for different periods, ranging from 2001 to 2021. For the current study, data from different years were used in the JULES model calibration (Table 1), selecting the most reliable set of observations, concerning data coverage and data quality assurance. The towers K34, K67, RJA and ATTO were used in model calibration. The tower K83 was used as an independent tower to validate the models for the spatialization developed in this study.



**Table 1: Description of five different eddy covariance towers based on Amazon region (Restrepo-Coupe et al., 2021).**

| ID | Site location Lat/Lon | Canopy height (m) | Year evaluated | Annual total rainfall (mm) | Average air temperature (ºC) |
|---|---|---|---|---|---|
| K34 | 2.614ºS/60.12ºW | 27 | 2005 | 1964.81 | 25.98 |
| K67 | 2.85ºS/54.97ºW | 36 | 2003 | 1283.72 | 28.19 |
| RJA | 10.08ºS/61.93ºW | 35 | 2001 | 2512.58 | 25.15 |
| K83 | 3.01ºS/54.88ºW | 28 | 2001 | 1658.29 | 26.37 |
| ATTO | 2.15ºS/59.03ºW | 30 | 2018 | 2192.87 | 26.03 |

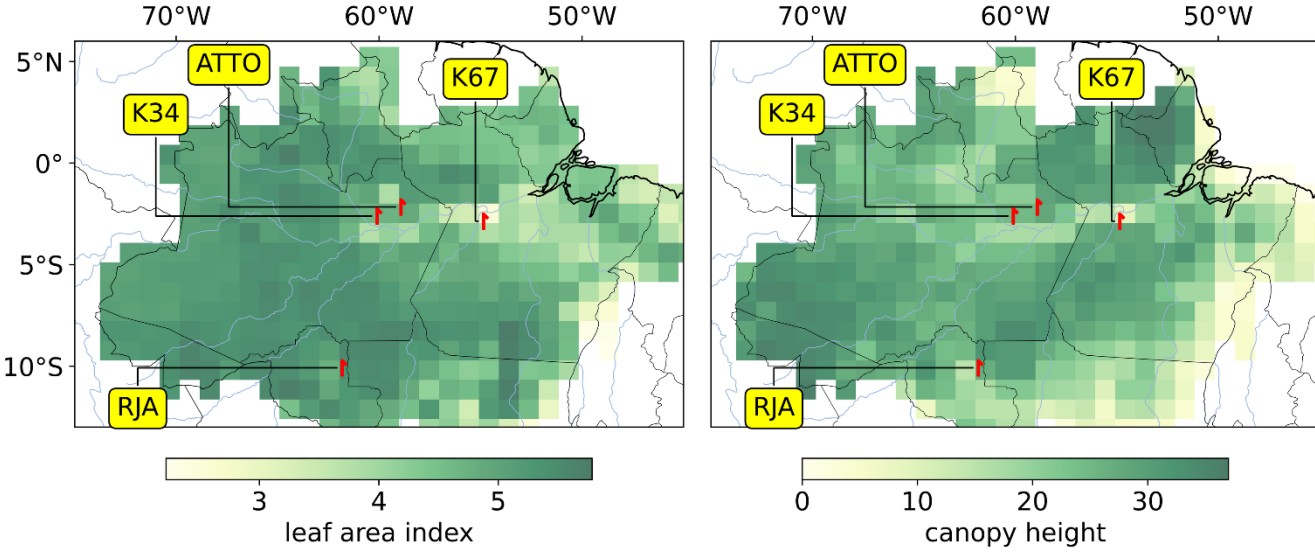

**Figure 1: Eddy covariance towers across the Brazilian Amazon biome (red symbols) used to validate JULES simulation. Grided background colors denote the spatial distribution of leaf area index ($m^2$ $m^{-2}$) on the left panel and canopy height (m) on the right panel (refer to section 2.3). The black symbol indicates the tower used to validate the spatialization of JULES parameters. LAI data is from ERA5 and Canopy Height data is from the Global Canopy Forest (Simardi et al., 2011).**

**2.2. Ancillary environmental data**



In-situ meteorological observational data were used as input to the JULES, while in situ carbon flux measurements were used as a reference to evaluate the model performance. JULES required a large number of meteorological data to run the model (Table 2). The reason is the complexity of approaching the photosynthetic process. Meteorological forcing data was derived from in situ observational data at each eddy flux tower site (Restrepo-Coupe et al., 2021, Andreae et al., 2015).

Soil information required by JULES was obtained from the EMBRAPA database (Reatto et al., 2004), which provides soil texture data (silt, sand, and clay content) at a 30 m of resolution down to a depth of 120 cm below surface. To convert soil texture into the parameters required to run JULES (Table 3) we applied equations from Marthews et al. (2014). The edaphological parameters in the model are static.

**Table 2: Meteorological variables required by JULES an their respective definitions and units.**

| Variable | Definition |
|---|---|
| sw_down | Downward flux of short-wave radiation, W m$^{-2}$ |
| lw_down | Downward flux of long-wave radiation, W m$^{-2}$ |
| Precip | Rainfall, kg m$^{-2}$ s$^{-1}$ |
| T | Air temperature, ˚C |
| Wind | Wind speed, m s$^{-1}$ |
| Pstar | Air pressure, Pa |
| Q | Specific humidity, kg kg$^{-1}$ |

**Table 3: Soil physical parameters required for JULES with their respective definitions and units for five different sites in the Amazon region.**

| Parameter | Definition | ATTO | K67 | K34 | K83 | RJA |
|---|---|---|---|---|---|---|
| b | Brooks-Corey exponential for hydraulic soil characteristics (dimensionless) | 15.65 | 11.20 | 15.65 | 11.20 | 6.52 |
| hcap | Dry heat capacity, J m$^{-3}$ k$^{-1}$ | 0.39 | 0.32 | 0.39 | 0.32 | 0.14 |
| sm_wilt | Soil moisture at the point of permanent wilt, m3 m–3 | 0.0006 | 0.0015 | 0.0006 | 0.0015 | 0.0066 |
| hcon | Dry thermal conductivity, W m$^{-1}$ k$^{-1}$ | 0.39 | 0.46 | 0.39 | 0.46 | 0.42 |
| sm_crit | Soil moisture at the critical point, m$^3$ m$^{-3}$ | 0.21 | 0.37 | 0.21 | 0.37 | 0.25 |
| satcon | Saturation hydraulic conductivity, kg m$^{-2}$ s$^{-1}$ | 0.12 | 0.26 | 0.12 | 0.26 | 0.14 |
| sathh | Soil matrix suction at saturation, m | 1236203 | 1228469 | 1236203 | 1228469 | 1272748 |
| sm_sat | Soil moisture at saturation, m$^3$ m$^{-3}$ | 0.20 | 0.22 | 0.20 | 0.22 | 0.27 |



| albsoil | Soil albedo (dimensionless) | 0.13 | 0.17 | 0.13 | 0.17 | 0.13 |
|---------|------------------------------|------|------|------|------|------|

Concerning the carbon fluxes, the variables utilized to calibrate and evaluate JULES simulations were NEE, GPP and Reco. It is important to mention that the direct observation from Eddy-Covariance tower measurement is NEE. NEE was partitioned following Botía et al (2022), who followed a similar approach as Restrepo-Coupe et al., assuming that nighttime NEE corresponds to nighttime Reco. Nighttime Reco was used as the daytime respiration, while daytime GPP was calculated from the difference between GPP and NEE (NEE=Reco-GPP). NEE data was available every 60 minutes for all flux towers,
except for the ATTO tower, available every 30 minutes.

## 2.3. Gridded data

    In addition to the in situ observational data, gridded datasets were used in JULES model simulations and as
benchmarks to the simulated carbon fluxes. Meteorological data from the ERA5 reanalysis (Hersbach et al., 2020) were used to force the JULES model in spatialized runs (refer to Section 2.4.3). ERA5 has hourly temporal resolution and a spatial resolution of 0.25ºx0.25º, which was resampled to 1ºx1º, providing data for the variables listed in Table 2.

    Gridded data of vegetation properties and land use were also used in the spatialized model runs, extrapolating model parameters across the Brazilian Amazon biome. Canopy height was collected in the Global Forest Canopy dataset (Simard et
al., 2011). This dataset represents the tree heights with a resolution of 927 m based on a fusion of spaceborne-lidar data (2005) from the Geoscience Laser Altimeter System (GLAS) and ancillary geospatial data. Canopy heights retrieved from the gridded product were similar to local observations at the 5 tower fluxes considered in this study.  LAI data from the ERA5 Land monthly reanalysis was used, with a resolution of 11132 m. In ERA5, LAI is calculated using the land surface model of the European Centre for Medium-Range Weather Forecasts, known as CTESSEL (Boussetta et al., 2013), with the assimilation of
a 9-year monthly climatology derived from satellite-based data from MODIS (Moderate Resolution Imaging Spectroradiometer). Therefore, the LAI product from ERA5 describes a fixed vegetation state. Land use and land cover data was provided by MapBiomas, collection 9, with a spatial resolution of 30 m (Souza Jr. et al., 2020). MapBiomas data were used to assign a PFT for each model grid (refer to supplementary material, Section 3.1, Table S3.1). All data was resampled to the 1ºx1º resolution, as described in Section 2.4.3.

Two gridded datasets on carbon fluxes were used as benchmarks for the simulations conducted in this study: FluxCom-X (Nelson et al., 2024) and the European Carbon Tracker CT 2022 (Jacobson et al., 2023). European Carbon Tracker provided hourly NEE at a resolution of 0.1° in latitude by 0.2° in longitude. To compare NEE with JULES simulation, the optimized biological flux was used (i.e., excluding Carbon flux from fuel and fire) and lateral fluxes from rivers were removed, following Friedlingstein et al. (2022).  FluxCom-X, provided with 0.05° spatial and hourly temporal resolution, is produced
using a data-driven approach using an ensemble of machine-learning methods, combining local observations from eddy





covariance flux towers, satellite observations and meteorological reanalysis data. In the Brazilian Amazon biome, FluxCom-X assimilates data from only two flux towers: K67 and K83. The scarcity of flux data in the Amazon hinders the model training, resulting in a decreased model performance in this region when compared to other terrestrial ecosystems worldwide. Overestimation of the carbon sink (strongly negative NEE) in tropical regions is a well-known bias of the FluxCom-X dataset

(Nelson et al., 2024).

## 2.4. JULES model

JULES is a land surface model that can simulate carbon fluxes punctually or in a grid with a temporal resolution of

one hour. The JULES version utilized in this study was 7.0, based on nine plant functional types (PFT), including tropical forests (Harper et al., 2016). JULES requires hourly meteorological data as input, as described in section 2.2. Also, it requires an edaphic dataset, which is also described in section 2.2. JULES estimates GPP and Reco based on the limitation factor of three potential photosynthesis rates (Collatz et al., 1991, 1992). A summary with a description of equations based on how JULES estimate carbon fluxes is demonstrated in the supplementary material S1.


### 2.4.1. Sensitivity analysis

The first step in process-based model calibration and local sensitivity analysis is to understand how the modulation of GPP and Reco is influenced by the model parameters (with NEE calculated by the difference of GPP and Reco). This study

initially assessed the sensitivity of the 21 core parameters of the JULES model by varying their values within the minimum and maximum expected ranges (Table S2.1). The underlying hypothesis was that the heterogeneity of the Amazon forest would lead to variation in these parameters. Understanding their impact on NEE helps identifying which parameters are critical for parameterization and should not be treated as fixed values, as it is done in the default JULES model PFT parameterization.

The local sensitivity analysis was developed for 2018 using the location of the ATTO tower as reference. Each JULES

parameter was perturbed within its maximum and minimum expected range, as shown in Supplement Table S2.1. The effect of these changes on NEE calculations was quantified using the mean absolute deviation (MAD, g C m$^{-2}$ day$^{-1}$) (Equation 1) and $\Delta$var (%) (Equation 2). MAD and $\Delta$var depend on the difference of NEE computed using the maximum and minimum value of a specific parameter (all others are maintained fixed with the default value). The calculation is computed for the simulation of each day and averaged over the year. $\Delta$var is computed as the sum of the square difference divided by the square

root of the number of days analyzed. To avoid outliers, we applied the Grubs test (Grubs, 1969) with a significance level of 95%, removing days with NEE considered outliers based on the absolute difference between maximum and minimum disturbed values, divided by the NEE before optimization (Harper et al., 2016). After this procedure, each parameter was classified by relevance level based on the largest $\Delta$var values, identifying the most sensitive parameters. Supplement Figures S2 present the NEE monthly simulations for 2018, considering the impact of changes in the most relevant parameters compared to observed





data (retrieved from the eddy covariance towers). The simulations were performed by varying each relevant parameter individually, using different values within the specified minimum and maximum range.

$$\text{MAD} = \frac{\sqrt{\sum_{i=1}^{N}(ymax_i - ymin_i)^2}}{N} \tag{1}$$

$$\Delta var\ (\%) = \frac{100}{N} \cdot \sqrt{\sum_{i=1}^{N} \frac{(ymax_i - ymin_i)^2}{(ydefault_i)^2}} \tag{2}$$

Where ymax is the NEE daily value simulated with the maximum parameter; ymin is NEE daily value simulated with the minimum parameter the minimum; ydefault is the daily value simulated using the default version of JULES default and N is the number of observations, removing days with outliers (352 days).

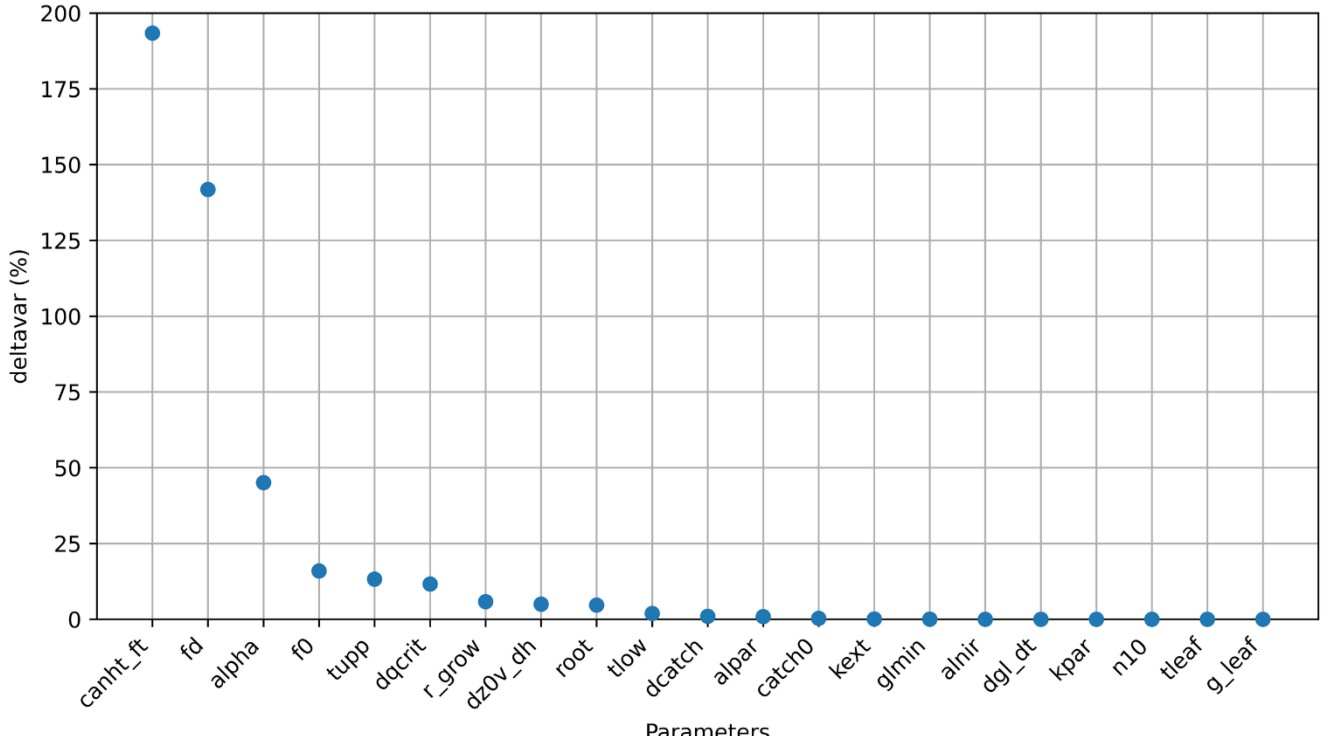

**Figure 2: Variation in (%) (Δvar) of JULES parameters relative to the default version of JULES at the ATTO tower, representing the Amazon biome during 2018. The abbreviations were defined in Supplement Table S2.1**

We considered the five most sensitive parameters (Figure 2): canopy height (canht); scale factor for dark respiration (fd); quantum efficiency (alpha); the maximum ratio of internal to external $CO_2$ (f0), and the upper-temperature threshold for photosynthesis (tupp). The sensitivity analysis has shown that variations in canopy height between 19 m and 50 m can lead to variations of almost 200% in NEE. The variation of the dark respiration scaling factor, for potential values found in Amazonia,



can also lead to differences in NEE of the order of 140%. The quantum efficiency, the maximum ratio of internal to external $CO_2$, and the upper temperature for photosynthesis can lead to variations in NEE of 45%, 16%, and 13%, respectively (refer to Table S2.2).

The set of parameters selected in the sensitivity analysis were similar compared to Raoult et al. (2016), which calibrated JULES for different plant functional types using GPP to evaluate the new parameterization. The reason behind the high sensitivity of NEE towards the parameter Canht can be explained by its influence on calculating the Maintenance Respiration (Equation 14, section S.1), being this parameter is necessary to estimate stem wood respiration (Clark et al., 2011). The parameter fd is also relevant to estimate the dark respiration coefficient (Equation 8, section S1) related to the Reco estimation. Alpha is a parameter that is related to estimating the Light-limited rate (Equation 5, section 2.1), f0 is a relevant parameter to estimate hydric stress and the stomata regulation (Equation 12, section 2.1) and tupp is required to estimate $V_{cmax}$ for different temperatures (Equation 1, section 2.1). The parameters alpha, tupp and f0 are highly related to calculate GPP being these three parameters also observed by Raoult et al. (2016) and Li et al. (2016) working on optimized GPP estimative using JULES. Optimizing this set of parameters makes it possible to spatialize the carbon fluxes in the Brazilian Amazon biome and their vegetation heterogeneity.

### 2.4.2. Calibration and validation

After the local sensitivity analysis, which defined the most important parameters for GPP and Reco, JULES was optimized comparing simulations with observed data at each site described in section 2.2. For this attempt, we used the Nelder-Mead method (Nelder and Mead, 1965) for the optimization, using the SciPy implementation (Harris, et al., 2020) and NumPy to process data (Virtanen et al., 2020). The Nelder-Mead method is a numerical method used to find the minimum or maximum of an objective function in a multidimensional space (Dakhlaoui, 2014). This method was successfully applied in studies of calibration and evaluation of different models as described by Jérôme et al., (2021). The JULES output utilized as a reference for calibration at each site was NEE, since it is directly retrieved from eddy covariance measurements without assumptions on flux partitioning. Canopy height was fixed as the average canopy height of each site, while the next four most sensitive parameters were concomitantly modified within the physiological limits, looking for the combination of values that minimized the error between model and observation on a daily scale (Table S4.1). After that, a leave-one-out cross-validation method was used to validate the calibration in different parts of the Brazilian Amazon biome (Wallach et al., 2018). The statistical index adopted to evaluate the error in this study was the root mean square error (RMSE) (Loague and Green, 1991, equation 3).

$$RMSE = \sqrt{\frac{1}{n}\sum_{i=1}^{n}(y_i - p_i)^2} \tag{3}$$

Where $y_i$ is the predicted value of NEE; $p_i$ the observed value of NEE and n the number of observations.

Another statistical matric used to analyze the accuracy of the simulations during the year was the index of agreement (*d*) proposed by Wilmott et al. (2012), given by



$$d = 1 - \left| \frac{\sum_{i=1}^{n}(yi-pi)^2}{\sum_{i=1}^{n}(|yi-P|+|pi-P|)^2} \right|, \tag{4}$$

where P is the average value of standard observations of NEE. When d is close to 1, this indicates a high accuracy level.

It was necessary to delimit maximum and minimum values for each parameter according to the physiological characteristics of tropical species (Table S4.1). Within the delimited values of reference, the optimization was developed from the default value of each JULES parameter adopted by Harper et al. (2016).

**2.4.3. Spatializing JULES in the Amazon biome**

The optimization of the JULES model parameters for different forest sites in Amazonia has shown significant differences, reflecting the heterogeneity of vegetation characteristics categorized in the PFT Broadleaf Evergreen Trees - tropical (BET-TR). This motivated the spatial extrapolation of JULES model parameters across the Amazon Basin using
remote sensing data as predictors.

After the sensitive parameters were adjusted for each tower, it was possible to spatialize the parameter values across the Amazon biome. As such, we have developed a spatially dependent parameterization of the BET-TR PFT in JULES. The spatialization model was based on linear regressions, having each sensible parameter as the target variable, and two remote-sensed vegetation properties as predictors: Canopy height and LAI. The reasons behind the choice of these variables include:
data availability and the fact that they can be used as a proxy for the 5 most sensitive parameters (Li et al., 2018; Moudry et al., 2024). Also, canopy height and LAI can be considered constant in the short term, describing a fixed vegetation state. Moreover, these two variables also were required as input for JULES simulations, canopy height as plant functional type parameter and LAI required for initial condition for simulations.

With independent linear regressions on canopy height and LAI, it was possible to extrapolate the model parameters
to the whole Brazilian Amazon biome. Different configurations for linear regression models were tested for each JULES parameter, following one of the general formats of Equation 5:

$$\begin{aligned} P(x,y,LAI) &= a + b.LAI(x,y) \\ P(x,y,height) &= a + b.height(x,y) \end{aligned} \tag{5}$$

where P represents JULES model parameters (tupp, alpha, f0_io and fd); x, y are the coordinates of each model grid cell; a, b, are regression coefficients to be determined. The regression models were fit to the parameters optimized at four forest sites
described in section 2.1 (K34, ATTO, K67 and RJA), using the maximum likelihood method. One of the towers (K83) was randomly left out, for means of validation. The choice of the regression model configuration for each JULES parameter was based on a compromise between the regression model residuals and the physical consistency of the extrapolated values. Section 3 in the supplementary material shows the reasoning behind the choice of each regression model.

To represent variations in carbon fluxes throughout the year, simulations were performed with one-degree resolution
across the Brazilian Amazon biome during April and September 2022, representing a wet and a dry season month in the





Amazon Region, respectively. The meteorological dataset required for JULES to simulate GPP, respiration, and NEE (Section 2.1, Table 2) was provided by ERA5 reanalysis data at an hourly scale with a 1º×1º resolution.

It is important to highlight that spatially dependent parameterization was used only for the BET-TR PFT, representing 71% of the Brazilian Amazon biome. For other PFTs present in the Amazon Basin, the default values were used for all
parameters (Harper et al., 2016). The canopy height for BET-TR was provided by our database utilized in this study and described in section 2.3; in relation to the other types of vegetation we used the values utilized by Clark et al. (2011) and Harper et al. (2016). In the case of C4 grass that has relevance in the arc of deforestation we utilized the canopy height of 15 cm, which is commonly used in the pasture present in the arc of deforestation (Fernandes et al., 2015). In the case of soybean, a relevant crop cultivated in the northern region of the state of Mato Grosso, we considered the sowing date in September and
the harvest in February, as described by Mato Grosso Institution of Agricultural Economics (De Lima Filho, 2021). To assign a PFT for each model grid, a correspondence was established between JULES land functional types and land use data from MapBiomas collection 9 (Souza Jr et al., 2020) (see supplementary material, Section 3.1, Table S3.1). Since MapBiomas data have a resolution of 30 m, it was necessary to calculate the percentual contribution of each land use class present in each 1º model cell grid (Figure S3.1). To run the model, it was necessary to introduce the fraction of each land functional type as a tile
to represent each vegetation present in the grid for JULES simulations. A description of all procedures utilized to spatialize JULES is described in Figure S4.5 (see supplementary materials section S.4).

## 2.5 VPRM model

The Vegetation Photosynthesis and Respiration Model (VPRM) (Mahadevan et al., 2008) is a satellite-driven empirical model designed to estimate NEE by integrating GPP and ecosystem respiration. GPP is calculated using a light-use efficiency method that combines meteorological inputs (e.g., temperature and photosynthetically active radiation) with remote sensing indices such as the Enhanced Vegetation Index (EVI) and the Land Surface Water Index (LSWI). These indices are derived from the MODIS Surface Reflectance 8-Day L3 Global 500 m (MOD09A1) product, collected within a ±0.1° area
around the ATTO tower. These data are interpolated to daily intervals using a curve smoothing technique (LOWESS filter). Ecosystem respiration is modeled using a linear function of temperature to capture the temperature dependence of carbon release. VPRM's key parameters include $\lambda 0$ (maximum light-use efficiency), PAR0 (light saturation constant), $\alpha$ and $\beta$ (coefficients controlling temperature dependence), as well as the temperature thresholds Tmin, Tmax, Topt, and Tlow. In this study, the parameter values employed were those calibrated by Botía et al. (2022) for the Amazon forest.

## 3.Results

## 3.1. Calibration and evaluation of JULES





After the identification of the model parameters with highest sensitivity in the Amazon, the JULES model was

calibrated for each flux tower, following the methods described in Section 2.4.2.  Table 4 shows the JULES default values for

the BET-TR PFT parameters (Harper et al., 2016) along with the optimized values considering local measurements in the

Amazon. The optimized values showed a strong variability, even among the equatorial forest sites. This explains the motivation

for the spatialization of JULES parameters for the BET-TR plant functional type.


**Table 4: New parameterization of JULES optimized by Nelder-Mead in each simulated site in this study. Four parameters were optimized: upper-temperature threshold for photosynthesis (tupp), quantum efficiency (alpha), scale factor for dark respiration (fd), maximum ratio of internal to external CO₂ (f0). Canopy height (canht) was retrived from observations at each site.**


| Parameter | unit | Default | ATTO | K67 | K34 | RJA |
|-----------|------|---------|------|-----|-----|-----|
| tupp | ºC | 43 | 42.18 | 36 | 42.77 | 36 |
| alpha | mol $CO_2$ per mol PAR photons | 0.08 | 0.05 | 0.066 | 0.061 | 0.05 |
| fd | dimensionless | 0.01 | 0.011 | 0.0066 | 0.01 | 0.007 |
| f0 | dimensionless | 0.875 | 0.95 | 0.713 | 0.93 | 0.875 |
| canht | m | 30 | 30 | 36 | 27 | 35 |

Figure 3 shows the simulated fluxes for GPP, Reco and NEE using optimized JULES parameters, JULES default

parameters, and simulations with the VPRM model. Observations are also depicted, as reference. The statistical metrics RMSE

and *d* (Equations 3 and 4) were calculated in each case, to assess the performance of each model setup in reproducing the

observations. The new parameterization reduced RMSE for GPP and NEE in all flux towers, in comparison to the default

parameter values and VPRM results. However, the optimized parameter values did not improve the Reco simulations. It is

important to note that NEE was the control variable in the calibration process so that the GPP and Reco partitioned fluxes were

indirectly optimized. The selection of the NEE as the control variable was related to the measured data that we have available

by Eddy-Covariance fluxes, and also to reduce the complexity in the calibration procedure due to the selection of five

sensitivity parameters that will influence in GPP and Reco.

The seasonality of the carbon fluxes was not captured by none of the model simulations. Flux measurements showed

an increase in GPP in the dry-to-wet season transition (Oct-Dec), while the models simulated either a weak annual cycle (RJA)

or a peak in GPP in the early dry season. The dry season effects were also described by Restrepo-Coupe et al. (2013) that



observed a different dynamic during the dry season in RJA tower, which is a region near to pasture and with a rainfall regime

different from the equatorial region represented by ATTO, K67 and K34 regions. In most process-based vegetation models, GPP is strongly associated with hydric stress, which may not be adequate for some Amazonian regions where leaf phenology and litter fall dynamics could play an important role (Restrepo-Coupe et al., 2017; Botia et al., 2022). None of the model simulations reproduced the observed carbon sink (NEE<0) between September and January at the K67 tower. Reco was overestimated in November and December, due to the directly relation between the dark respiration coefficient (fd) and $Vc_{max}$

(Clark et al., 2011, equation 8 section S1). Despite the limitations in the reproduction of the carbon fluxes seasonality, the optimization of JULES parameters resulted in improved estimates for annual means in NEE, reducing the bias in comparison to the default parameter values.

VPRM demonstrated weaknesses in simulating GPP seasonality, and the error magnitude in NEE was higher than in the optimized JULES and, in some regions (K67 and K34), even higher than its default version. Botia et al. (2022), comparing

different models in ATTO tower, reported that VPRM demonstrated low efficiency in capturing carbon seasonality in this region. This was attributed to the lack of methods for calculating hydric stress, as the only water scaling source was the Water Scale Index, derived from remotely sensed Land Surface Water Index using MODIS reflectance data (Chandrasekar et al., 2010 and Gourdji et al., 2022).









**Figure 3: GPP, Reco, and NEE simulations using different model setups and observations at each flux tower in Amazonia. The observed data in the plots is the accumulated during each month of the year, and the RMSE error described is the daily average during the year in g C m$^{-2}$ day$^{-1}$.**


### 3.2. Spatialization of JULES parameters

Considering the variability of the optimized parameters for different sites of the Brazilian Amazon biome (Table 5), simple linear regression models were developed to extrapolate the parameter values for the whole Brazilian Amazon. As predictors, vegetation characteristics described in the Section 2.4.3, namely LAI and canopy height, were used. Table 5 shows the linear regression models developed for the JULES parameter with the highest sensitivity. The reasoning behind each regression model is available in section 3 of the supplementary material, where maps showing the spatialized values for the JULES parameters and the relationship of each parameter with the respective vegetation index are also shown.

The relationship between each parameter and the selected predictor is shown in Figure 4. Canopy height was selected for tupp and alpha, while LAI was selected for f0 and fd. Tupp showed an inversely proportional relationship with the canopy height (Figure S.3.2.2), which is consistent with that fact that low-canopy plants like C4 grasses typically have higher temperature thresholds for photosynthesis. The parameter alpha did not show a clear relationship with any of the predictors, resulting in a rather constant behavior against canopy heigh (Figure S.3.3.2). Canopy height was chosen as a predictor for alpha to obtain the expected lower quantum yields for C3 and C4 plants (0.055 mol[1] mol$^{-1}$, Skilman 2008) (Figure S3.3.1). Parameter f0 was positively associated with LAI (Figure S.3.4.2), consistent with the fact that f0 is expected to be lower in the arc of deforestation compared to forest sites. For fd parameter, the selected predictor was LAI (Figure S.3.5.2), which is expected to have a positive relationship with fd, given the greater photorespiration efficiency in C4 plants.









**Figure 4: Spatialization of the parameters tupp (upper-temperature threshold for photosynthesis), alpha (quantum efficiency), f0 (maximum ratio of internal to external CO₂), and fd (scale factor for dark respiration) for the Amazon biome using two different methods: based on canopy height and based on LAI.**

The regression equations were used to obtain the parameter values at the K83 tower site, which was left aside in the spatialization process. Using a canopy height of 28 m and an average LAI value of 3.78, as described in section 2.3, the parameter values obtained for K83 were used in a JULES simulation for the year of 2001, obtaining the GPP, Reco and NEE fluxes depicted in Figure 5.

**Table 5: Parameterization based on the spatialization in the Amazon region for four JULES parameters in Tower K83**

| Parameter | Equation | $R^2$ | Value extrapolated to K83 |
|---|---|---|---|
| Tupp (°C) | 66.6212-0.85574*Height | 0.94 | 42.66 |
| alpha (mol mol$^{-1}$) | 0.04942 + 0.00022*Height | 0.01 | 0.056 |
| f0 | 0.40266 + 0.10577*LAI | 0.87 | 0.802 |
| fd | -0.00101+0.0022*LAI | 0.91 | 0.0073 |

The parameterization proposed for JULES was used to simulate GPP, Reco, and NEE at the K83 tower, compared against observations in Figure 5. The most relevant aspect was the improvement in GPP reducing the RMSE in 37% in comparison to the default version of JULES and 39 % in comparison to the VPRM model. Observations at the K83 tower showed a weak annual cycle in the carbon fluxes, which was satisfactorily reproduced by the models. Overall, this validation process indicates that the method used for the spatialization of JULES parameters provided satisfactory estimates in a forest site that was left out of the regression models.





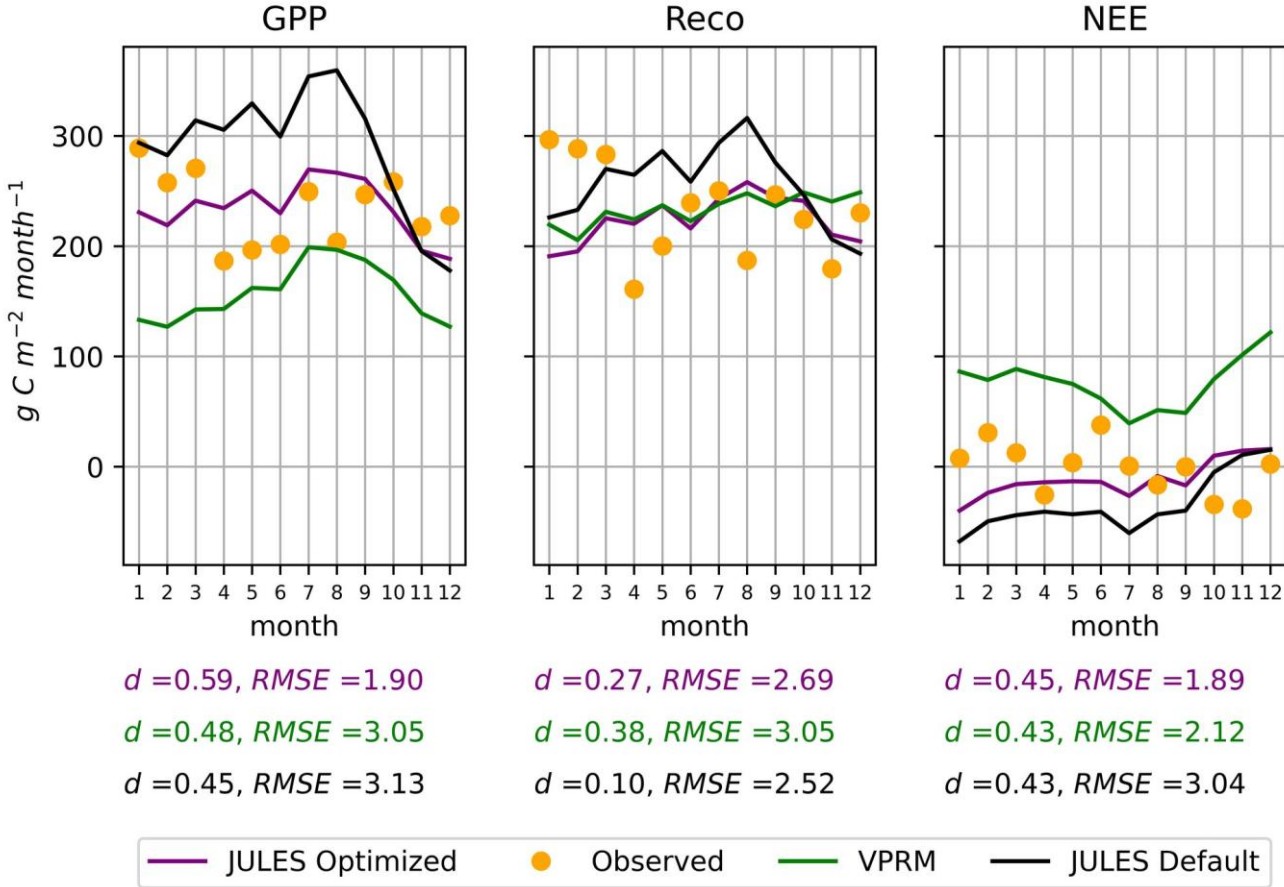

**Figure 5: GPP, Reco, and NEE fluxes in the independent Tower of validation K83, for the year of 2001.The observed data in the plots is the accumulated during each month of the year, and the RMSE error described is the daily average during the year in g C m⁻² day⁻¹.**

## 3.3. Spatial and seasonal variability of carbon fluxes in Amazonia

After validation with an independent tower (K83), we were confident in using JULES to estimate carbon fluxes across the entire Brazilian Amazon biome for the year 2021. This year was chosen to allow comparison of the simulated carbon fluxes with recently released or updated global datasets. The simulations used the spatialized values of the 5 most sensitive parameters of the BET-TR JULES PFT (Fig. 4). Default parameter values were used for other PFTs in the Amazon Basin. Figure 6 highlights the results from two representative months of the wet (April) and dry (September) seasons in Amazonia.



The mean GPP in April was 223 g C m$^{-2}$ month$^{-1}$, while the mean Reco was 170 g C m$^{-2}$ month$^{-1}$, characterizing a

carbon sink (NEE) of -53 g C m$^{-2}$ month$^{-1}$. During the dry season month (September), there was an increase in GPP, reaching

a mean of 240 g C m$^{-2}$ month$^{-1}$, in Reco, with a mean of 182 g C m$^{-2}$ month$^{-1}$, increasing the carbon sink to -58 g C m$^{-2}$ month$^{-1}$. The fact that GPP was not reduced by water limitation during the dry season also was observed by Restrepo-Coupe et al.

(2013) who observed an increase in GPP during the dry season based on observations at the flux towers K34, K67 and K83.

The Reco value estimates by JULES are underestimated in the wet season April when compared to Botia et al. (2022),

which reported a mean value of 350 g C m$^{-2}$ month$^{-1}$ for the wet season at the ATTO tower. In the dry season, however, Reco

estimates were similar to the values reported by Botia et al. (2022) (200 g C m$^{-2}$ month$^{-1}$) in the same region, which suggests

that further improvements are needed to better reproduce the seasonality of Reco in Amazonia, particularly in the Amazon

basin.

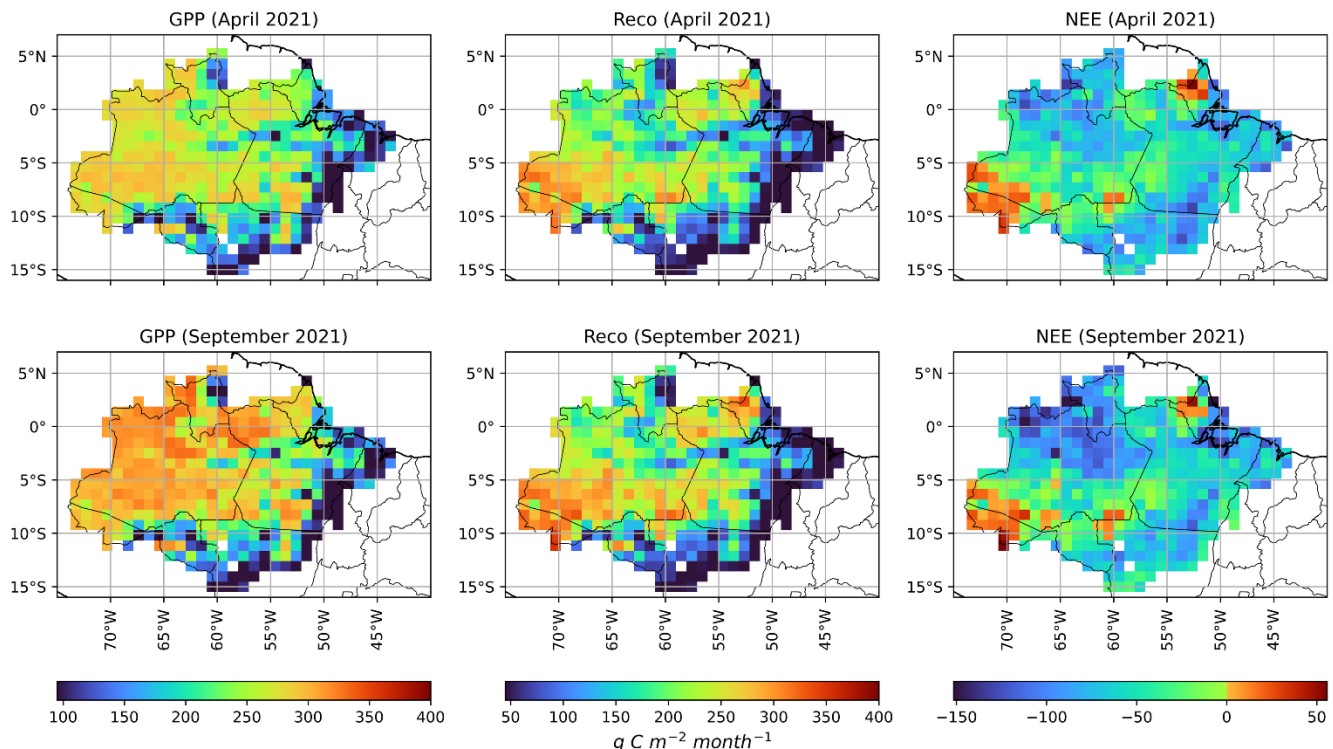


**Figure 6: Monthly accumulated GPP, Reco, and NEE for April and September, representing the wet and dry season in the Brazilian Amazon biome in 2021.**

To further evaluate the spatialized model results, the simulated NEE fluxes for the year of 2021 were compared to the

following estimates (Figure 7): i) European Carbon Tracker CT 2022 (Jacobson et al., 2023); ii) FluxCom-X (Nelson et al.,





2024); iii) JULES simulation using fixed average values for the BET-TR parameters, considering the optimized values presented in Table 2; iv) JULES simulation using default parameter values (Harper et al., 2016).

Figure 7 clearly shows that the three different modelling approaches using JULES (optimized, default, and spatially fixed best adjusted parameters) result in an overestimation of the carbon sink in Amazonia (i. e., more negative NEE values) during both the wet and dry seasons, when compared to Carbon Tracker and FluxCom-X. The JULES run with spatialized vegetation parameters reveals spatial structures in the NEE fluxes, such as the less intense carbon sink observed in the far western Amazonian region (Acre state). There are no carbon flux measurements in this region to be used as ground truth, but the optimized Carbon Tracker estimates also indicate less intense carbon sink in this area, or even a carbon source (NEE>0)

in the wet season (Figure 7). Another spatial pattern revealed in the JULES simulation with spatialized vegetation parameters was a reduced carbon sink in northern Amazonia (Amapa state and northern Para state), where the Carbon Tracker and the FluxCom-X datasets also detected reduced carbon sinks or even carbon sources.

The regions of Acre and Amapa regions demonstrated a high carbon source. What these regions have in common are canopy heights above the average, with trees reaching above 35 m in Amapa (Figure 1). Compared to FluxCom-X, JULES

simulated a stronger carbon sink across the Brazilian Amazon biome during both the wet and dry months, except in forests located in the states of Amapa and Acre (Figure 7). This feature was also observed in the average JULES version and can be attributed to a modification of the tupp parameter in the BET-TR (43 ºC). Restrepo-Coupe et al. (2017) simulated a reduction of GPP during the dry season, however the version used in this study was the 2.1, based on the parameterization of Clark et al. (2011), which considered a tupp of 36 ºC in comparison of our study that utilized a tupp between 36 to 45ºC as a limit in the

calibration procedure. Also, another relevant aspect that may have induced the GPP increases in JULES's simulations was the higher values of f0 observed in some regions calibrated in this study, such as ATTO and K34 (0.95 and 0.93, respectively). The modification of f0 led to a reduction in water stress, mainly in areas of the Amazon associated with the Amazon basin and the northern region of the State of Mato Grosso. Moreover, f0 may also help explain the carbon source in the states of Amapá and Acre, as the calibration procedure indicated an f0 value near 0.7 in regions with more sparsely spaced trees (LAI <4.0),

which can contribute to an increase of Reco, as shown in Figure 3.

Although JULES simulations highly overestimated the carbon sink, they showed a similar trend compared to the Carbon Tracker estimates, with an increase in magnitude of the carbon sink from April to September. The states of Acre and Amapa showed similar patterns to Carbon Tracker, both representing a reduced carbon sink in this areas (less than 50 g C m$^{-2}$ month$^{-1}$) (Figure 7). This similarity can be explained by the effects of the spatialization of JULES parameters mainly in regions

with tall trees as the case of forests of Amapa and Acre. During the wet month, Carbon Tracker showed a larger carbon source area across the Amazon biome compared to FluxCom-X, which represented a carbon source mainly in April in Amapá, in some regions of Amazon basin, the state of Acre, and the deforested areas of Roraima. (Figure 7). During September, Carbon Tracker was similar to FluxCom-X in representing a carbon sink across most of the Brazilian Amazon biome. However, FluxCom-X showed a carbon source in the arc of deforestation, which was not indicated by either JULES or Carbon Tracker.





Another important aspect was that the JULES spatialized leads to a weaker sink of carbon in NEE in comparison to the default

and mean versions (Figure 7).

**Figure 7: Comparison of NEE fluxes for April (wet season) and September (dry season) of 2021 using different modeling approaches:**
**JULES model with spatialized vegetation parameters (spatialized JULES); European Carbon Tracker, FluxCom-X, JULES model using the spatially constant mean values of optimized vegetation parameters (mean JULES), and JULES model using default vegetation parameter values (default JULES). Differences between spatialized JULES and the other estimates are also presented.**





After applying the spatialization procedure and comparing with different models, the JULES model was run for the
entire year of 2021. Monthly accumulated NEE values were summed up from each 1º x 1º pixel to estimate NEE for the
Brazilian Amazon biome, resulting in -1.34 Pg C year $^{-1}$. It is important to mention that this value represents the sum of
different regions within the Amazon biome (Figure 8). The one-year accumulation revealed that the most concentrated carbon
sources are located in the states of Amapa and Acre, with values exceeding 0.75 x 10$^{-12}$ Pg C m$^{-2}$ year $^{-1}$ of carbon released to
the atmosphere. Some regions stood out as strong carbon sinks (below -1.0 x 10$^{-12}$ Pg C m$^{-2}$ year $^{-1}$) as the forest in the north
of the State of Mato Grosso (longitude 53ºW and latitude 12ºS) and the forest of São Gabriel da Cachoeira (longitude 74ºW
and latitude 0º N). The forests in the Amazon basin also demonstrated a high carbon sink (-0.25 to 1.0 x 10$^{-12}$ Pg C m$^{-2}$ year $^{-1}$) during 2021.

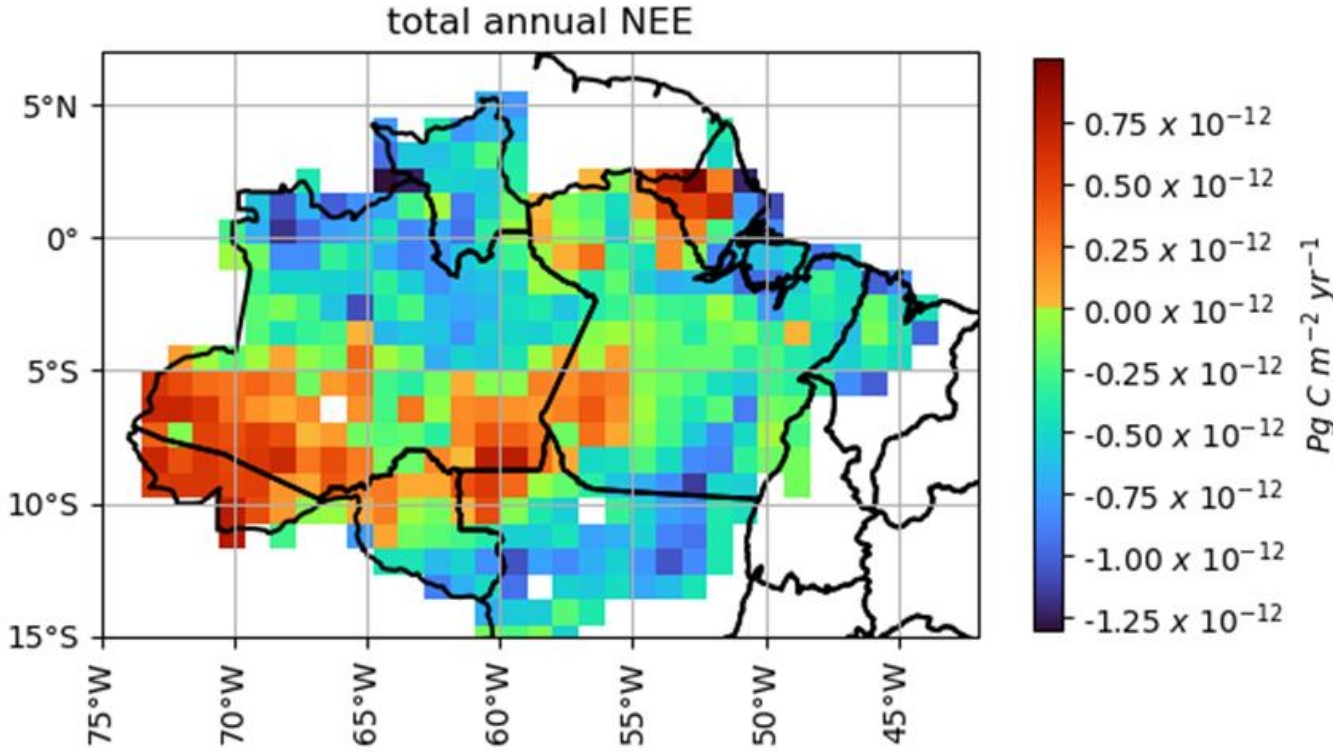

**Figure 8: NEE accumulated in Pg C m$^{-2}$ during 2021 in the Brazilian Amazon biome.**

**4. Discussion**

**4.1. JULES optimization**






The optimization of JULES for four parameters at the four flux tower sites in this study showed convergence in their values, consistent with the findings of Clark et al. (2011) for tupp value (36 ºC) for Santarém and Jarú, and the f0 value for Jarú (0.875). Additionally, the f0 value calibrated for Santarém in this study was close to that reported by Raoult et al. (2016), which was 0.765. The tupp values for ATTO and K34 were similar to those reported by Harper et al. (2016), who increased

tupp to 43 ºC for tropical forests. The parameter fd was similar to the value reported by Harper et al. (2016) for ATTO and K34, however, it was lower than 0.01 for Santarém and Jarú, with values of 0.0066 and 0.007, respectively. The alpha parameter was lower than that in all previous JULES calibration studies for each of the towers. However, the results of this study for alpha parameter are in line with the values near 0.05 and 0.06 mol PAR mol $CO_2^{-1}$ reported by Skilmann (2008), who analyzed different types of $C_3$ plants, including Broadleaf Evergreen Trees. This could be explained by the way that the default

version overestimated GPP in comparison to the calibrated version (Figure 4).

The cross-validation procedure also showed that the alpha parameter was more variable than the other parameters (Figure S4.2). This may be explained by the fact that the values for each tower were nearly constant, with a difference of only 0.0016 mol PAR mol $CO_2^{-1}$, while the other parameters exhibited greater variability between sites. The spatialized JULES also showed lower RMSE for NEE compared to VPRM model. This can be explained by the greater complexity of how JULES estimates

GPP and, particularly, Reco. VPRM uses a simpler approach, relying on a linear regression in which air temperature is the sole independent variable (Gourdji et al., 2022). In contrast, JULES estimate Reco with more sophisticate equations that account for factors such as water stress and the nitrogen content in different plant components (Best et al., 2011; Clark et al., 2011).

## 4.2. NEE estimates using JULES spatialized


The first relevant aspect that spatialized JULES was able to reproduce was the increase of GPP during the dry season, showing that water may not be a limitation for carbon assimilation in the Amazonian dry season (Figure 3 and Figure 6). Restrepo-Coupe et al. (2013) also observed the same feature by comparing different Eddy-Covariance Towers spread in the

Brazilian Amazon biome. The main driver for seasonality in this region is Canopy Photosynthetic Capacity and the leaves phenology (Restrepo-Coupe et al., 2013). The use of different parameterizations in different sites across the Amazon biome, with different LAI values, may have helped capture the effect of leaf phenology, particularly the emergence of new leaves, which tend to open their stomata for photosynthesis more frequently than older leaves approaching senescence (Wu et al., 2016). The absence of phenology representation in some process-based models was noted by Restrepo-Coupe et al. (2017) and

Botia et al. (2022), who observed a tendency for these models to underestimate GPP during the dry season. In contrast, the spatialization method used in this study, incorporating varying parameters based on LAI and canopy height, improved the model's ability to simulate GPP and Reco with greater diversity (Figure 6).

The spatialized JULES generated values of NEE between 0.75 and -1.25 x $10^{-12}$ Pg C $m^{-2}$ $year^{-1}$. This range is according to the observed by Lian et al. (2023), which estimated an average value of NEE in the South America Forest of -0.205 Pg C



m$^{-2}$ year$^{-1}$ using a Randon Forest Model applied in a global system. The spatialized JULES also demonstrated that the major focus of carbon sources was located in the states of Amapa and Acre (Figure 7). The forest of Amapa has the characteristic of having tall trees (> 35 m) and a low leaf area index (< 4.5 m$^2$ m$^{-2}$) even considering that this region has the major above-ground biomass of the Amazon biome, reaching 518 Mg ha$^{-1}$ (Ometto et al., 2023). The spatialization reduced the tupp and f0, which may lead to a reduction in GPP in relation to Reco, generating a carbon source in this region. A possible explanation is increased

Reco in mature and tall forests, such as those found in the Amapa region. West (2020) observed in a review of studies with different tree species that the costs of respiration increase over the years while GPP remains constant, which could explain the net carbon source. The costs are based on adjustments in morphology and anatomy to construct new structures in the xylem, roots, and leaves to support the high amount of biomass. Concerning the state of Acre there is a climatic condition that can explain the carbon source in this region, since annual rainfall is lower than 2000 mm (Silva et al., 2020), which can increase

the cost of maintenance and hence the respiration to avoid hydric stress. It is important to have eddy-covariance measurements in this region, to confirm the trend of carbon source.

        The total NEE estimated for the Brazilian Amazon biome demonstrated a carbon sink (NEE) of  -1.34 Pg C year $^{-1.}$ The result obtained in this study is between the results obtained by Chen et al. (2024), which estimated the NEE in the Amazon region using the Trendy-v11 (-0.94 Pg C year$^{-1}$) and FluxCom-RS (-3.46 Pg C year$^{-1}$) on an annual average between 2001-

2015. It is important to mention that the FluxCom version utilized to analyze the NEE across the Brazilian Amazon biome (Figure 7) is a new version with the X-BASE database (Nelson et al., 2024) in comparison to the FluxCom-RS utilized by Chen et al. (2024). The new version of FluxCom reduced the global NEE in relation to the FluxCom-RS (-21 to -7 Pg C year$^{-1}$). Trendy v-11 is a dataset that provides global gridded carbon fluxes data from 16 different types of vegetation dynamics models (Sitch et al., 2022). Our result was closer to the Trendy than to the FluxCom-RS, which can be favorable considering

the uncertainties in the NEE partitioning in FluxCom-RS, being the Reco partitioned by the NEE instead of subtracting GPP to get NEE (Jung et al., 2020).  Due to this reason, FluxCom-RS underestimated Reco and tends to estimate a higher carbon sink. JULES is included in Trendy v-11, however, the version utilized to simulate carbon fluxes was JULES 5.1 (Wiltshire et al., 2021). In this version, JULES has only five Plant Function Types and the version we utilized to simulate carbon flux is v7.0. In this version, we have the parametrization specific for the Tropical Evergreen Broadleaf tree (Harper et al., 2016). One

sensitivity parameter modified in this version was the tupp (36°C in the Clark et al 2011 to 43°C in the Harper et al., 2016). This could be a reason that the carbon sink in the Brazilian Amazon biome is larger than the JULES version in TRENDY, being this parameter relevant for GPP. Due to this reason, in the section S.5 of supplementary material, we compared the default with the spatialized version and the version utilized in Trendy v-11 based on Clark et al 2011 parametrization which demonstrate that the Harper et al., (2016) version tends to overestimate the carbon sink and the spatialized JULES version

approximate with observed data in ATTO tower for 2018 (Figure S5).

        The result of -1.34 Pg C year$^{-1}$ was lower than found by  other studies related to the Amazon region as Botia et al (2024) (-0.33 Pg C year$^{-1}$) and Rosan et al (2024) (-0.34 Pg C year$^{-1}$), however, some aspects need to be considered. The first aspect is related to the number of years evaluated in these studies in relation to our study. We have just evaluated the year 2021





instead of other studies that evaluated more than ten years, as the case of Chen et al (2024) and nine years as the case Botia et al (2024). The second aspect is that the carbon flux obtained by Botia et al (2024) (Net Land Flux) is the sum between river fluxes and NEE; in our simulation, we have just the NEE obtained by vegetation, similar than the paper by Chen et al (2024) which generate a NEE of -0.94 Pg C year$^{-1}$ using the Trendy-v11 and 3.46 Pg C year$^{-1}$. The third aspect is related to the fire emissions that can contribute to reducing the carbon sink, this value can vary 0.09 Pg C year$^{-1}$ (Rosan et al., 2024) to 0.41 Pg C year$^{-1}$ (Gatti et al., 2021).

Another important aspect to be mentioned and that can contribute to this distance between other models is that some process-based models can overestimate the carbon sink in tropical forests, as previously related by Restrepo-Coupe et al., (2017) and also by Botia et al. (2022) mainly when compared with inversion models as Carbon Tracker. The reason can be explained by the incorrect assumption of water limitation and the lack of leaf phenology in model formulations (Gonçalves et al., 2020). Also, JULES demonstrated a higher sink in other types of vegetation presented in the Amazon biome as C4 grass (Harper et al., 2016) and C3 crops (Williams et al., 2017; Prudente Junior et al., 2022) in regions such as the states of Mato Grosso, Roraima, and east of Pará (Figure 8). In a region predominantly composed by C4 grass (longitude 49.5°W, latitude 7.5°S), with 84.5 % of C4 grass (see supplementary material, Figure S3.1.1), JULES simulated a carbon sink of -250 g C m$^{-2}$ year$^{-1}$ (Figure 8). This value is higher (more negative) than that reported by Bezerra et al., (2022), which obtained in eddy-Covariance tower a NEE annual mean of -215 g C m$^{-2}$ year$^{-1}$ in the Brazilian Northeast, working with *Urochloa brizantha* cv Marandu, the most relevant pasture used in the arc of deforestation. This indicates that tropical grassland can be considered a carbon sink mainly in regions with latitude near 15° S to 0°, with similar radiation levels during different seasons of the year. However, it is important to point out that the improvement of grassland parameterization is out of the scope of the current study. One step that can improve the estimate of NEE in the arc of deforestation can be the calibration and evaluation in agricultural crops, which can reduce the carbon sink in the regions of Mato Grosso and Roraima.

The spatialized JULES demonstrated a higher carbon sink in comparison to FluxCom-X and the European Carbon Tracker in the Amazon basin, deforestation arc, and North of Mato Grosso, although it presented a higher carbon source in the states of Acre and Amapá. However, the new optimization and the spatialization approach showed improvements over the version used by Harper et al. (2016), which applied averaged optimized parameters. In addition to reducing the estimated carbon sink, it also highlighted the influence of vegetation heterogeneity on the spatial distribution of carbon budget across the Amazon biome, particularly in the states of Amapá and Acre.

## 5. Conclusions

This study presented a new method to estimate NEE from the adjusted land surface model and spatialization considering two relevant vegetation properties: Canopy height and LAI. The first aspect presented in this study was to demonstrate the most sensitive parameters for NEE, being the canht, tupp, alpha, f0 and fd the most important parameters for the parameterization procedure. The optimization of selected JULES parameters for the PFT BET-TR led to a reduction in both



RMSE and the d-index across all four analyzed towers, when compared to the default parameter values and the VPRM model. Our attempt of spatialization was validated in an independent tower, generating a better performance than VPRM and the
default version of JULES. In general, the spatialized JULES model showed a stronger carbon sink in the northern Amazon region and across the Amazon basin compared to FluxCom-X and Carbon Tracker, particularly during the month of September. However, the spatialized version of JULES also indicated significant carbon source regions (> 75 g C m$^{-2}$ month$^{-1}$) in Amapá and Acre. This highlights the importance of considering how forests with tall canopy height (>35m), such as those in Amapá, and the influence of climate conditions, as observed in Acre, contribute to the overall carbon budget. The spatialized JULES
resulted in a NEE estimative of -1.34 Pg C during 2021, which is a value that approaches those with dynamic vegetation models for the Amazon biome.

Despite the advances presented in this study, some aspects still need more robust explanations. One of these aspects is related to the strong carbon source in the regions of Amapá and Acre, being the simulations obtained by JULES in 2021 a relevant aspect to better investigate the carbon balance in these regions. It is important to note that this study developed an
optimization of JULES using a limited number of eddy Covariance towers. While this approach improved model simulations, further improvements could be achieved by installing additional towers in different forest types across the Amazon region, especially in regions with tall canopy heights (> 35 m). Another aspect that could be improved is the simulation of regions dominated by agricultural land uses, such as soybean, maize, and pasture. These areas, particularly in northern Mato Grosso, are relevant because the model currently simulates them as carbon sinks. Despite the development of JULES-crop, this model
is not coupled in the most recent version of JULES, a feature that could improve simulations in agricultural zones. Additionally, calibration and evaluation using Eddy covariance towers in croplands and pastures could improve model performance in the deforestation arc. Despite these limitations, this study highlights the relevance of spatializing NEE using vegetation indices, demonstrating how this approach can improve the mapping of carbon fluxes in the Brazilian Amazon biome by identifying source and sink regions in relation to forest height and density.

**Funding Information**

Funding sources include the support of the RCGI – Research Centre for Greenhouse Gas Innovation hosted by the University of São Paulo and sponsored by FAPESP – São Paulo Research Foundation (process number 2020/15230-5). FAPESP also funds the projects 2023/06623-1, 2022/07974-0, 2023/04358-9 and 2024/12950-8. Also, the CNPq - National
Council for Scientific and Technological Development to fund the project 304819/2022-0.

**Data and code availability**

Modelling data is available under request.

**Author contributions**



ACPJ wrote the initial manuscript and ran the JULES model for the Brazilian Amazon biome. ACPJ, together with LATM, LVR, SB, and FSS, designed the methodology. DSM assisted in adapting JULES for the Amazon region. LPC ran VPRM model for different sites across the Brazilian Amazon biome. CQDJ provided meteorological data measured in the ATTO tower. LATM, LVR, PEAN, TA and EF provided Amanan's computers to run JULES. LATM, CP, SB and PEAN consolidated

funding for the postdoctoral position and an exchange period at the Max Planck Institute for Biogeochemistry in Jena. LATM, LVR, XX, SB helped with the data curation and the interpretation of the results. IMCT helped to improve observed carbon fluxes at different sites in the Amazon region. FSS contributed to developing scripts to run JULES and to designed figures presented in the manuscript. LATM, LVRM, XX, SB, FSS, EF, CQDJ and IVCT contributed to review the manuscript.

**Competing interests**

The authors declare that they have no conflict to interest.

**Acknowledgment**

The authors acknowledge the use of the Amanan's Clusters, a collaborative facility supplied under the Institute of
Astronomy, Geophysics and Atmosphere Science of the University of São Paulo. Also, the Max Planck Institute of Biogeochemistry for the ATTO towers data availability and the São Paulo Research Foundation to fund the project 2023/06623-1.

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
