# Peer review of "Spatializing Net Ecosystem Exchange in the Brazilian Amazon biome using the JULES model and vegetation properties"

_EGUsphere, 2025_

## Author Comment (AC2)

**Responses to Reviewer**

We would like to thank the reviewers for their valuable comments and useful suggestions to improve our manuscript. Below are the responses and actions for each comment. To facilitate the identification of individual responses and actions, we have employed the following colour coding strategy. In black are the reviewer's comments. In blue are the author's responses, and in blue italic the new text added to the manuscript.

- The study only uses single years, with little discussion about why this was done or how useful it is (eg, how large is NEE interannual variability). Presumably, these flux sites have data for more years, so why was this not used?

Thank you for raising this question. We certainly need to clarify this aspect in the manuscript. The primary objective of this study is to propose a strategy for spatializing JULES parameters, given that they remain constant in the standard JULES model (they are default values for all regions in the namelist). Rather than providing the NEE for the Amazon, we intend to evaluate the effect of spatialization on the Brazilian Amazon biome. The data coverage in the LBA-ORNL dataset varies greatly across the five flux towers considered in this study. Some towers have a couple of years of intermittent valid measurements (e.g. RJA and K34), while others have several years of quality-assured data (e.g. ATTO and K67). In particular, measurements of within-canopy $CO_2$ storage were intermittent in some of the towers, and this variable is crucial for ensuring unbiased NEE estimates. Consequently, we selected a single year with high-quality data from each tower to provide similar conditions for model parameter optimization at the different flux sites. We recognize that there is interannual variability in NEE, particularly during droughts (e.g. Botia et al., 2022). To avoid the influence of atypical years on model optimization, years affected by El Niño were avoided. To clarify this point, we have rephrased a couple of sentences in section 2.1.

*"Some of these flux towers are still operational, while others have been discontinued. As such, observations from each tower are available for different periods ranging from 2001 to 2021, sometimes with intermittent measurements." For the current study, data from*

*different years were used in the JULES model calibration (see Table 1), with a complete year being selected that had the most reliable set of observations in terms of both data coverage and quality assurance. Using a single year of data for each site provided similar conditions for model parameter optimization at the different sites. To minimize the influence of atypical conditions reflected in the variability of carbon fluxes, years with extreme dryness or wetness were avoided during the model optimization process. "*

- The organisation of Section 2 is difficult to follow. It would be clearer to put JULES model description (S2.4) before JULES model inputs and ancillary data (S2.2) and JULES model forcing data (S2.3). The resulting description of the model the paper is very short (six sentences and no equations). The authors should bring some relevant information out of the SI and into the main paper. For example, the paragraph at L209 mentions several equations that should be in the main paper, particularly showing how maintenance respiration depends on canopy height, which isn't obvious from Eq 14 in the SI.

We would like to thank you for this suggestion, and we have improved section 2 accordingly. We added a description of the main JULES equations for the calculation of GPP and Reco. Also, we introduced the equation of Shinozaki (1964) describing the respiring stemwood and the relationship between canopy height and stem carbon content. We also reorganized the order of the subsections, following the reviewer's suggestion .

- The authors never mention model initialization or soil moisture spin-up: how was this done? At some points is sounds like the model was run separately for individual months (eg L270) and a full year run was made after the shorter runs (eg L459 "After... the model was run for the entire year"). If so, why was this done and how was the model initialized? The manuscript would benefit from a table describing the set-up for simulations run in this study (eg duration, forcing, parameter source).

We would like to thank you about this suggestion. Despite the relevance of the spin-up technique, in this study we did not adopted this procedure due to the computationally

expensive and would be impractical for in the case of this study that we have hundreds of grids points. To reduce the need for spin-up, we considered the strategy adopted by Moreira et al (2013) that utilized to run JULES in the Amazon region and considered to run the model from the start to the end of the simulation period. Regarding soil moisture, we considered the EMBRAPA database (as described in the section 2.3) provided values that is near from the observed data of soil texture, this can reduce uncertainties in relation of the water balance model. In relation to the carbon pool, we not altered during the simulation, and carbon levels varied in accordance with seasonal changes throughout the year. We added the following text in the section 2.5 in JULES procedures.

*"Attaining equilibrium between carbon stocks and humidity via the soil moisture spin-up procedure was a computationally expensive process. For this study, it was difficult to implement because of the large number of grid points required to simulate the Brazilian Amazon region. To initialise the JULES simulations, we adopted the strategy employed by Moreira et al. (2013). We ran JULES from the start to the end of the simulation period. The carbon pool was not altered during the simulation, and carbon levels varied in accordance with seasonal changes throughout the year. Also, we considered the soil texture obtained in the EMBRAPA database (described in the section 2.3) as a source that closely matches with the observed data and this can reduce the uncertainties in the water balance "*

- The authors only quote the annual total NEE for the run using new parameters (-1.34 PgC/y) and not for the runs with default or mean parameters. This seems like an odd omission given that the default parameter run forms a baseline for comparison with other values (eg, TRENDY). I feel that those annual results should be reported and discussed.

We thank you for this suggestion, and we ran JULES for the entire Brazilian Amazon biome in the year of 2021 using the parameters of the default version by Harper et al (2016) and the spatial mean values of the optimized parameters proposed in this study. JULES default estimate a carbon sink of -3.08 Pg C year$^{-1}$, and the version utilizing the mean of the most sensitive parameters optimized reduced the carbon sink to -2.06 Pg C year$^{-1}$. This demonstrates that JULES spatialized reduced the carbon sink in the Brazilian Amazon biome in 56.49 % in relation to the default version and in 34.96 % in relation to

the mean version. Figure A shows NEE simulated using the default, mean and spatialized version of JULES. The spatialized version of JULES resulted in a greater spatial variability in NEE, mainly in areas with annual rainfall lower than 2000 mm, as in the case of the south of the Amazonas and Acre state. In these regions, JULES spatialized simulated a carbon source above $0.75\,\text{kg C m}^{-2}\,\text{year}^{-1}$, and in the mean and default version, these regions demonstrated a carbon source between 0 to $0.25\,\text{kg C m}^{-2}\,\text{year}^{-1}$. It shows that the spatialized version was able to better represent the water stress, following the approach to spatialize two parameters directly associated with hydric restriction (fd and f0).

[Figure]

**Figure A: NEE accumulated in kg C m⁻² during 2021 in the Brazilian Amazon biome in default, mean and spatialized version.**

The Figure demonstrated in this letter was added separately in the supplementary materials as Figure S5.2 (default version) and Figure S5.3 (mean version). A paragraph regarding the comparison between JULES default, mean and spatialized was added in the text in the section 4.2.

*"In comparison with the annual value obtained by the mean and default versions of JULES (Harper et al., 2016), the default version obtained a carbon sink of -3.08 Pg C per year (see Supplementary Material, Figure S5.2), while the mean version obtained a carbon sink of -2.06 Pg C per year (see Supplementary Material, Figure S5.3). The default version of JULES presented a value similar to that obtained by FluxCom-RS (-*

*3.46 Pg C per year), demonstrating that the calibration procedure adopted in this study improved the carbon simulations by JULES despite the lack of FluxCom-RS equations to simulate Reco. Another piece of evidence demonstrating the improvements made by the calibration procedure is that the mean value of the optimised parameters reduced the carbon sink in the Brazilian Amazon biome by 33.12% compared to the default value. The spatialised version of JULES reduced the carbon sink of the Brazilian Amazon biome by 56.49% compared to the default version and by 34.96% compared to the mean version, reaching a value closest to that provided by Trendy-v11 (-0.94 Pg C year⁻¹) by Chen et al. (2024). This reduction in the carbon sink can mainly be explained by the regions of Acre, as shown in Figures S5.2 and S5.3 for the default and mean versions, respectively. This can be considered the effect of the method of spatialising the sensitivity parameters f0 and fd, which are directly related to water stress (Clark et al., 2011), as characterised in this region. The same aspect can explain why the spatialised version of JULES demonstrated a high carbon source in the south of the Amazonian state (>0.50 kg C m⁻² year⁻¹), which the default and mean parametrizations did not capture (between 0 and 0.25 kg C m⁻² year⁻¹). However, it is worth noting that the state of Amapá demonstrates a carbon source in all three versions of JULES, reaching 0.75 kg C m⁻² year⁻¹. This suggests that the height of the tree canopy in this region contributes to the carbon source."*

- I'm disappointed to see no uncertainties reported in this paper, as they are really required for comparing points estimates. For example, the authors omit the uncertainties when quoting from Chen et al (2024) and from the calculation of their headline NEE value of -1.34 PgC/y. Similarly, I would expect to see uncertainty estimates on the fitted parameter values in Table 4.

We clearly understand your suggestion to report uncertainties in this study. Effectively, every study providing NEE for Amazonas has no reference to rely on. This is an important point that we tried to quantify during the study; however, there are no precise values, despite the few eddy-covariance stations. Even the in-situ data have disagreement compare LBA-ECO with FLUXNET, for the same station, we found disagreement,

mainly based on the way they interpolate the data, and they consider the storage flux and compute respiration. We computed NEE by using the approach of Botía et al (2022), who followed a similar approach as Restrepo-Coupe et al., assuming that nighttime NEE corresponds to nighttime Reco. The only real measurement we can rely is the in-situ station, and we adjusted the model to these variables and spatialized the JULES parameters using the Nelder-Mead method of optimization. This method does not generate uncertainties for the fitted parameters. We selected a minimum and maximum value based on physiological limits of the plant as described in the Table S4.1, and the method selected a value in this range, without demonstrating uncertainty in a confidence level. We introduce in the text a sentence clarifying this limitation in section 2.5.2.

*"Important to mention that the Nelder-Mead method does not generate uncertainties for fitted parameters at a confidence level, being limited to one value in a physiological range that will be our reference to the calibration procedure."*

Regarding the headline value that we found (-1.34 Pg C year$^{-1}$) we do not intend to have these values as the NEE for Amazonia, because they correspond to only one year and there are uncertainties related to the adjustments and the database employed (tree height and LAI). We were not able, due to the computational resources available, to run JULES with uncertainties to provide this sensitive test. However, we have the simulation using the default JULES mode, and the average values of the new parameters to compare with the main simulation. We have added this discussion to the text in section 4.2.

"*The result of -1.34 Pg C per year can be analysed as the result of the spatialisation procedure for 2021 and cannot be considered the absolute value of net ecosystem exchange for the Amazon. The uncertainties of this value can be evaluated by comparing it with the default estimation for the same year, which provides a much larger value. Calculating the Amazon net ecosystem exchange requires the use of different years (El Niño and La Niña years), a more precise ancillary database, and, of course, more eddy covariance stations covering different Brazilian Amazon biomes."*

- Data and code availability: the results here depend on many JULES input options that it would not make sense to report in the manuscript, but which should be made

available to readers. The minimum I expect these days is for the input namelist files to be made available to readers via Zenodo or similar.

Thank you for the suggestion, we introduced the dataset covering all simulations described and JULES namelists at this link: http://ftp.lfa.if.usp.br/ftp/public/LFA_Processed_Data/articles_database/Prudente_2025/. Also we added a new sentence in the Data and code availability section.

"*The dataset covering all simulations described in this report is available at this link:http://ftp.lfa.if.usp.br/ftp/public/LFA_Processed_Data/articles_database/Prudente_2025/.* "

- Generally, the manuscript needs more work for clarity and readability, and typographical mistakes need fixing, particularly in the SI (eg, PFT changes to LFT, changing signs on NEE values, Sitch et al 2022 should be 2024).

We would like to thank you for this suggestion, and we corrected some typographical mistakes in the new version as well as the comments of the other comments.

Other comments

· L58 "The model includes up to nine land cover types containing five PFTs". JULES can have any number of tiles/PFTs, but common configurations are five PFTs (HadGEM3), nine PFTs (Harper et al 2016), or 13 PFTs (UKESM1). This line is also inconsistent with the model description in S2.4 (L165) that mentions nine PFTs being used in this study.

Thank you for the suggestion, the version that we utilized to simulate NEE using JULES was v.7.0, in this version has 9 PFTs and 4 non PFT. We introduced in the lines 58- 60 this consideration regarding configurations of PFT in JULES.

*"The model includes different configurations of plant functional types (PFT): five PFTs (HadGEM3, Clark et al., 2012), nine PFTs (Harper et al 2016), or 13 PFTs (UKESM1, Harper et al., 2018)"*

· L97: "The tower K83 was used...". Only later in the paper (L266) is it mentioned that K38 was chosen "at random". The reason for choosing K83 should be mentioned here in the methods S2.1.

Thank you for the suggestion, and we replaced the criteria to select the K83 tower as independent for validation in the section of study area 2.1.

*"The tower K83 was used as an independent tower to validate the models for the spatialization developed in this study. Tower K83 was left out, for means of validation."*

· Table 3: Four of the sites have sm_crit > sm_sat, which is very unusual and possibly inconsistent with other model assumptions. With JULES sm_crit and sm_wilt are usually diagnosed at standard hydraulic pressures (-33 and -1500 kPa respectively) from sathh, b, sm_sat and the hydraulic equation being used. Could the authors explain this apparent discrepancy?

Thank you for this observation, and in fact there is a discrepancy between sm_wilt and sm_crit and others errors. This was a typing error but the values retracting the edaphological data used for simulations in JULES was provided in the new version of Table 3.

· L137: Why did the authors choose to resample from 0.25 to 1.0 degree? The former is a common resolution of land surface modelling (eg ISIMIP) and is inexpensive to compute for a limited region for single years.

We worked with data at various resolutions, including ERA5 at 0.25°, MapBiomas land use data at 30 meters, and ERA5 Land data at 0.1°. One important aspect is that our simulations was made for all Brazilian Amazon biome and this procedure serves to provide the downscale of WRF-GHG with CMIP (Coupled Model Intercomparison Project) data

in the resolution of 1x1º. In view of the future use of this method and the computional limitations to run throughout the Brazilian Amazon biome, we selected this resolution. In view of clarification, we introduce a sentence explaining the resolution of 0.1º in the section 2.4

*"This resolution was proposed in view of the computational limitations to run JULES for the Brazilian Amazon biome. "*

· L148: "...used to assign a PFT for each model grid...": This description isn't clear. I think from reading later in the paper that the LAI values were simply assigned to the BET-Tr PFT

We utilized the Mapbiomas data as reference to represent the land use in each grid. The question regarding the spatialized method for sensitivity parameters of JULES was used only for Broadleaf Evergreen Tropical Trees (BET-TR) taking into account that the regression models was based only in areas with 100 % of Tropical Trees. However, in areas with agricultural crops or others land use types, we utilized the parameters of Harper et al., (2016). We rephrase the sentence in the section 2.4.

*"MapBiomas data was the reference to run JULES for each PFT represented in each grid (refer to supplementary material, Section 3.1, Table S3.1). All data was resampled to the 1ºx1º resolution and utilized in different versions of models approached in this study, as described in Section 2.5.3 "*

· L183: "...fixed with the default values": The default values for the 21 parameter used in the sensitivity analysis should be reported.

We would like to thank you for this suggestion and we introduced a plot describing the NEE simulated in the ATTO tower during the year of 2018 using the parameters of Harper et al., (2016). The plot is in the supplementary material as Figure S2.2 and described in the section 2.5.1.

*"ydefault is the daily value simulated using the default version of JULES default (Figure S2.2)"*

· L185: "Grub's test": Isn't Grubb's test (note the spelling) used to detect outliers from random errors, such as spurious observations? A model doesn't have random errors, so the authors should describe what conditions they are attempting to catch with this test.

We deeply appreciate the notice regarding the misspelling of Grubbs's test and the reference (Grubbs, 1969). Your keen eye for detail is invaluable to us. We will incorporate these corrections in the revised version of the text, ensuring the accuracy and credibility of our work. The $\Delta$var (%) values include the expression $(ydefault\_i)^2$ in the denominator of a fraction. When these values are close to zero, $\Delta$var (%) can reach significantly higher magnitudes than the other values, leading to a loss of meaning at this point. To address this issue, we utilized Grubbs's test to identify and exclude these spurious data from our sets of $\Delta$var (%) values. In view of clarification in the text, we added to the text in the section 2.5.1.

*"Δvar is computed as the sum of the square difference divided by the square root of the number of days analyzed which can generate spurious values with significantly higher magnitudes. To mitigate the impact of these spurious values, we treated them as outliers and applied the Grubbs' test (Grubbs, 1969) with a significance level of 95%, removing days with NEE considered outliers based on the absolute difference between maximum and minimum disturbed values, divided by the NEE before optimization (Harper et al., 2016)"*

· L193: Eq 1 for MAD: This is not the common definition of mean absolute deviation, which is SUM(ABS(ymax_i-ymin_i))/N. The equation that's written describes a "mean root sum of square deviation", so it's not the RMSD either. Is there a typo in this equation?

We would like to thank for this observation and we replaced the mean absolute deviation to mean rot sum of square deviation during the section 2.5.1 also in the Table S2.2 in the Supplementary materials.

*" NEE calculations was quantified using the mean root sum of square deviation (RMSD, g C m$^{-2}$ day$^{-1}$) (Equation 13)"*

·    L211: "...high sensitivity of NEE": Another reason for the sensitivity could be because the roughness length is also linearly related to canopy height, which will affect the carbon fluxes.  Can the authors rule out this as a significant factor in the sensitivity?

In fact, besides the Maintenance Respiration is calculated by JULES using canopy height, the roughness length can also explain the high sensitivity of NEE for canopy height. Best et al., (2011) described that JULES calculates the roughness length for momentum based on the canopy height and rate of change of roughness length with vegetation canopy height which is one parameter of JULES that varies for different plant function type and land use. The effect of roughness length in carbon fluxes can be explained by the mechanical turbulence and the capacity to enhance the mixing of air and to facilitate the transfer of gases, including $CO_2$, between the land and the atmosphere (Khanna and Medvigy, 2014). We introduced this aspect in the section 2.5.1.

*"Another relevant aspect that explain the high sensitivity of Canht is the linear relation with roughness length (Best et al., 2011), which is important for carbon fluxes estimative by the mechanical turbulence and the capacity to enhance the mixing of air and to facilitate the transfer of gases, including $CO_2$, between the land and the atmosphere (Khanna and Medvigy, 2014)"*

·    L277: "C4 grass... canopy height": This is a confusing detail to include as no other information about C4 grass is included, and it makes it sound like the authors used the grass height to derive parameters from Eq 5, which I understand was only used for the BET-Tr PFT.  Could the authors clarify?

We fixed the canopy height for $C_4$ grass in 15 cm due to the value utilized in the default version for grassland is 1.26 m and for $C_3$ grass is 0.76 (Harper et al., 2016). If we simulate JULES using these values canopy height, the carbon sink should be overestimated as the high biomass that these crops would be able to accumulate. Taking into account that

Brazilian pasture in the arc of deforestation is widely used for cattle feed, the value utilized for farmers of cattle entrance in the grassland of *Urochloa Brizantha* cv Marandu is 15 cm.

*"This is necessary in view of avoid overestimative in the carbon sink in this region taking into account that is a option to maintain the grassland in  a height typical for the catlle farms of this region"*

- L326: "...not captured by none...": Accidental double negative?

We would like to thank you for this observation and replaced this sentence to "*The seasonality of the carbon fluxes was better represented by JULES optimized utilizing the Nelder-Mead method (Figure 3).*

- L359-367: Is the comparison with C4 grasses in this paragraph relevant to the parameterization fits that are specifically for BET-Tr trees? The fits are over tree heights between 27 and 36 m, so extrapolating down to a different vegetation class with heights of 0.15 m seems to me a poor comparison and not very meaningful.

We recognize this limitation, however, it is important to mention that we have a limited number of Eddy-Covariance available in the Brazilian Amazon biome which that was not found a specific tower to measure C4 pasture in the deforestation arc. The extrapolation to other plant functional type was an alternative to understand the dynamic of JULES main sensitive parameters and a comparison with other studies using JULES for different plant functional type was our reference to validate the vegetation property selected for the regression model. Other strategy was to evaluate the highest $R^2$ which means that had a linear relation with Canopy Height or LAI. Despite the Canopy height present a low value of $R^2$ for alpha, the histogram presented in the Figure 4 showed that the range of simulation of this parameter in the deforestation arc is in line that observed by Skillman (2008) (0.05 – 0.06 $mol^1$ $mol^{-1}$ for Tropical Trees), which gave some confidence to spatialize this parameter. We introduced this topic in the section 3.2.

*"We recognise that these regression models are limited in their ability to represent alpha estimators, despite the values obtained for the Brazilian Amazon biome being in line with those of Skilman (2008).*

- Table 5 and Figure 5: I note that the K83 parameter values are similar to the default values from Table 4 (possibly with the exception of alpha). Presumably that means most of the improvement at K83 in Figure 5 is because of the canopy height value directly, which was prescribed from satellite data. Could the authors elaborate on how much of the model improvement was because of the parameters in Table 5 rather than the prescribed values of canopy height and LAI?

We need to clarify that the canopy height in the tower K83 is the same that was utilized in the spatialized version, taking into account that LAI and Canopy Height were two parameters that was not presented in the regression models because they were collected directly from Global Forest Canopy dataset and ERA 5. Our intention with the validation tower was to test the regressions models developed for four parameters that were estimated with this methodology. In view of clarifying this aspect, we introduced that Canopy Height and LAI values were the same utilized in the Default, Spatialized and VPRM model in the final of the section 2.4.

*"All data was resampled to the 1° x 1° resolution and utilized in different versions of models approached in this study, as described in Section 2.5.3"*

- L418: "Table 2": Presumably the authors mean different table?

We would like to thank you for this observation and replaced "Table 2" to "Table 4".

- L470: Figure 8: Couldn't "10^-12 Pg C" be simplified to "kg C"? I understand the wish to keep it consistent with other uses of Pg C in the paper, but those are usually area totals, which are not easily compared with these per unit area values anyway.

We would like to thank you for this observation and replaced the unit Pg C m$^{-2}$ to Kg C m$^{-2}$ in the Figure 8.

- L492: "...water stress and nitrogen...": Perhaps more importantly, JULES accounts for factors such as radiation and humidity, which are strongly connected with the alpha an f0 parameters, respectively, that the authors show are influential. Could the authors comment on this aspect too?

We would like to thank you for this suggestion and we introduced a sentence about this aspect in the section 4.1. In fact, the aspect that Reco estimate by JULES has more complexity in view of equations that takes into account water stress and nitrogen, the GPP and Reco also performed better than the JULES default version because we spatialized alpha and f0 parameters which improved GPP and Reco simulations. The description added in the section 4.1. is described:

*"In contrast, JULES estimate GPP and Reco with more sophisticate equations that account for factors such as water stress, nitrogen content in different plant components (Best et al., 2011; Clark et al., 2011) also utilizing equations that can define the energy utilized for photosynthesis as the light-limited rate (equation 3) and the leaf concentration of $CO_2$ based on the leaf humidity deficit (equation 5) including in this aspect two sensitivity parameters: alpha and f0, that were modified in the spatialized version and can explain the best performance comparing the default version and VPRM model."*

- L508: "-0.205 Pg C m-2 yr-1": This should be kg rather than Pg. Also, isn't is unsurprising that the mean NEE for a region (from Lian et al) lies roughly in the middle of the range of extremes of spatially resolved values from this study?

We would like to thank you about this observation and we replaced -0.205 Pg C m$^{-2}$ year$^{-1}$ to -0.205 kg C m$^{-2}$ year$^{-1}$. Regarding the question about the mean NEE found in the study of Lian et al (2023) confirms that the range of values that JULES spatialized generate can be approximated to the mean value observed by Lian et al (2023), however, the study realized by the authors did not spatialized to verify zones of sink or source of carbon as our methodology proposed. Due to this fact, the value observed by Lian et al(2013) serves as reference for our

range in JULES spatialized. We introduced a sentence describing this aspect on the section 4.2.

*"The spatialized JULES generated values of NEE between 0.75 and -1.25 Kg C $m^{-2}$ year$^{-1}$. This range is according to the observed by Lian et al. (2023), which estimated an average value of NEE in the South America Forest of -0.205 Kg C $m^{-2}$ year$^{-1}$ using a Randon Forest Model applied in a global system. In view of the spatialization procedure adopted in this study, the mean value obtained by Lian et al (2023) in the Amazon biome serves as a appropriated reference for our range spatialized. "*"

References

Best, M. J., Pryor, M., Clark, D. B., Rooney, G. G., Essery, R. L. H., Ménard, C. B., Edwards, J. M., Hendry, M. A., Porson, A., Gedney, N., Mercado, L. M., Sitch, S., Blyth, E., Boucher, O., Cox, P. M., Grimmond, C. S. B., and Harding, R. J. (2011). The Joint UK Land Environment Simulator (JULES), model description–Part 1: Energy and water fluxes. Geoscientific Model Development, 4(1), 677–699. https://doi.org/10.5194/gmd-4-677-2011, 2011.

Botía, S., Komiya, S., Marshall, J., Koch, T., Gałkowski, M., Lavric, J., and Gerbig, C.: The CO2 record at the Amazon Tall Tower Observatory: A new opportunity to study processes on seasonal and inter-annual scales. Global Change Biology, 28(2), 588-611. https://doi.org/10.1111/gcb.15905, 2022.

Clark, D. B., Mercado, L. M., Sitch, S., Jones, C. D., Gedney, N., Best, M. J., Pryor, M., Rooney, G. G., Essery, R. L. H., Blyth, E., Boucher, O., Harding, R. J., Huntingford, C., and Cox, P. M.: The Joint UK Land Environment Simulator (JULES), model description – Part 2: Carbon fluxes and vegetation dynamics, Geosci. Model Dev., 4, 701–722, https://doi.org/10.5194/gmd-4-701-2011, 2011.

Grubbs, F. E: Procedures for detecting outlying observations in samples. Technometrics, 11(1), 1-21. https://doi.org/10.1080/00401706.1969.10490657, 1969.

Harper, A. B., Cox, P. M., Friedlingstein, P., Wiltshire, A. J., Jones, C. D., Sitch, S., and Bodegom, P. V.: Improved representation of plant functional types and physiology in the Joint UK Land Environment Simulator (JULES v4. 2) using plant trait information. Geoscientific Model Development, 9(7), 2415-2440. https://doi.org/10.5194/gmd-9-2415-2016, 2015.

Harper, A. B., Wiltshire, A. J., Cox, P. M., Friedlingstein, P., Jones, C. D., Mercado, L. M., ... & Duran-Rojas, C. (2018). Vegetation distribution and terrestrial carbon cycle in

a carbon cycle configuration of JULES4. 6 with new plant functional types. *Geoscientific Model Development*, *11*(7), 2857-2873. https://doi.org/10.5194/gmd-11-2857-2018

Khanna, J., & Medvigy, D. (2014). Strong control of surface roughness variations on the simulated dry season regional atmospheric response to contemporary deforestation in Rondônia, Brazil. *Journal of Geophysical Research: Atmospheres*, *119*(23), 13-067.

https://doi.org/10.1002/2014JD022278

Lian, Y., Li, H., Renyang, Q., Liu, L., Dong, J., Liu, X., and Zhang, H.: Mapping the net ecosystem exchange of CO2 of global terrestrial systems. International Journal of Applied Earth Observation and Geoinformation, 116, 103176. https://doi.org/10.1016/j.jag.2022.103176, 2023.

Moreira, D. S., Freitas, S. R., Bonatti, J. P., Mercado, L. M., Rosário, N. E., Longo, K. M., & Gatti, L. V. : Coupling between the JULES land-surface scheme and the CCATT-BRAMS atmospheric chemistry model (JULES-CCATT-BRAMS1. 0): applications to numerical weather forecasting and the CO2 budget in South America. Geoscientific Model Development, 6(4), 1243-1259. https://doi:10.5194/gmd-6-1243-2013, 2013.

Shinozaki, K., Yoda, K., Hozumi, K., and Kira, T.: A quantitative analysis of plant form – the pipe model theory, I. Basic Analyses, Japanese Journal of Ecology, 14, 97–105, 1964

---

## Author Comment (AC3)

**Responses to Reviewer**

We would like to thank the reviewer for the valuable comments and useful suggestions to improve our manuscript. Below are the responses and actions for each comment. To facilitate the identification of individual responses and actions, we have employed the following colour coding strategy. In black are the reviewer's comments. In blue are the author's responses, and in blue italic the new text added to the manuscript.

Main comments

First, the authors acknowledge that the availability of eddy covariance data is limited. Still, a more comprehensive assessment of how representative these sites are of the region would make it easier to see if/where extrapolations go beyond the training data. For example, because canopy height and LAI products are used to predict model parameters, it would be useful to know to what extent these four sites cover the range of variation seen across the Amazon.

The five flux towers considered in this study sit in upland forest sites, in regions subjected to subtle differences in the climate regime and stress associated with land use change. We acknowledge that the flux towers are not representative of all Amazonian ecosystems, like seasonally flooded forests. Even though upland forests represent more than 80% of the Amazon Biome, and the range of canopy height and LAI values at the 5 towers comprises most of the spatial variability of these quantities (Figure 1 in the manuscript and Figure A below). In a grid of 1-degree resolution, upland forests predominate everywhere, except in the Arc of Deforestation, where pasture-land use dominates. As such, the extrapolation from the flux tower sites to the domain of upland forests is reasonable.

[Figure]

Figure A: Histogram of canopy height and LAI in the study area. The range of heights and LAI in the 5 flux towers considered in this study is highlighted in red.

To clarify this point, we have included in Table 1 the LAI values corresponding to each flux tower. We also included the following paragraph in section 2.1:

*"All the eddy covariance flux towers are located in upland (terra firme) forest sites, with canopy heights in the range 27-36 m and LAI in the range 3.26-5.46 $m^2$ $m^{-2}$ (Table 1), respectively comprising 61% and 85% of the distribution of values in the study area (Figure 1). The five flux towers considered in this study represent upland forests in regions with subtle differences in the climate regime and in the level of stress associated with deforestation and climate change pressures. While the K34 and ATTO towers are located in pristine forest reserves in the Western Amazonia, the RJA tower sits in a forest reserve surrounded by agricultural areas in Southwestern Amazonia. The K67 and K83 towers sit in a forest reserve near the deforestation frontier in the Eastern Amazonia. Therefore, this set of flux towers represents the diversity of upland forests, which extend through more than 80% of the Amazon biome (Moraes et al., 2021). However, it is important to acknowledge that this set of flux towers is not representative of all Amazonian ecosystems, like seasonally flooded, swamp or white sand forests.   "*

Second, given that the parameter tuning at the eddy covariance sites did not reproduce observed seasonal flux patterns (stated in line 326 and shown in Figure 4), I was surprised that seasonal patterns were so widely highlighted in subsequent Amazon-wide simulations (Figures 6, 7, discussion lines 496-507). The earlier results did not seem to support that JULES is appropriate for investigating seasonal patterns.

Indeed, our results have shown that the model calibration did not improve the representation of the NEE seasonal variability, although it did improve the mean bias. Even so, to better understand the model response to different meteorological forcings, we believe it is important to assess the NEE spatial variability in periods with contrasting atmospheric conditions. To comply with the reviewer's comment, in the discussion of Figure 6, we have acknowledged the poor representation of the NEE seasonal variability, including the following sentence in the section 3.3:

*"Despite the fact that the JULES model was not able to precisely reproduce the carbon flux seasonal cycles (Fig. 3), it is important to assess the estimated spatial variability of NEE in months with contrasting meteorological conditions, investigating the model responses."*

We also provided a new title in the section 3.3. to "Spatial variability of carbon fluxes in Amazonia" and we selected two months in two distinct seasons in view of to understand how the meteorological effects change the spatialization in the Brazilian Amazon biome. We introduced this sentence in the section 3.3.

*"Evaluating JULES over two months enables us to understand the dynamics of spatialisation in two distinct seasons. Despite JULES' limitations in reproducing seasonality, this approach is useful for determining spatial variability in dry and wet conditions."*

Last, it was not clear to me whether the K83 validation exercise was conducted with in-situ meteorological data or the ERA5 meteorological data. It would be helpful to whether using ERA5 data decreases data-model agreement for all eddy covariance sites.

Thank you for raising this point. In situ meteorological data was used for both the calibration and validation of the model at the K83 tower. ERA5 meteorological data was only used for the specialization of carbon fluxes. When calibrating a process-based model, we considered that the meteorological data needs to demonstrate a high level of confidence; in this case, in-situ data is recommended (Wallach, 2019). To clarify this in the manuscript, we have reworded the sentences in section 2.3 as follows:

*'The JULES model requires the meteorological variables listed in Table 2 as input. In-situ meteorological forcing data from each flux tower (Restrepo-Coupe et al., 2021; Andreae et al., 2015) were used for model calibration (Section 2.4.2) and cross-validation using K83 tower data. For the specialization of carbon fluxes, meteorological data from reanalysis was applied, as will be described in Section 2.4.'*

Specific comments

- Lines 55-57: For a general audience, I think it would be helpful to describe JULES more broadly. For example, I see that JULES can be run either as a "big-leaf" or a "multi-layer" approach. From the supplement I see that the multi-layer implementation was used here, and I think a couple sentences about this should also be included in the main text.

Thank you for the suggestion. We have added a description of JULES containing equations that highlight where the most sensitive parameters are, as well as describing how JULES simulates GPP and Reco. We also introduced information on our use of the multi-layer approach to run JULES. The new organisation of the material and methods involves the JULES description in section 2.2; section 2.3 covers ancillary environmental data; section 2.4 focuses on gridded data; and section 2.5 describes the JULES model procedures and the spatialisation method. The final section provides a brief description of the VPRM model.

- Lines 139-142: I am somewhat surprised that GEDI-based products were not used for this task, as GEDI's sampling is now denser in this area (https://doi.org/10.3334/ORNLDAAC/2339). It would also be helpful to see what the Global Forest Canopy predictions are for the eddy covariate sites, perhaps in the supplement. It is somewhat hard to tell from Figure 1.

We appreciate the suggestion regarding the database used for canopy height. In view of this, we compared GEDI's sampling with the Global Forest Canopy dataset (Figure B).

Despite the high positive correlation (0.7) between these two datasets, differences can be observed mainly in the reproduction of canopy heights between 5 and 10 meters.

However, the aim of our study is not to provide the most up-to-date NEE for the Brazilian Amazon biome, but to present a methodology for JULES using spatialized main sensitivity parameters.

Of course, the results are sensitive to the assimilated dataset. We used the Global Forest Canopy dataset because it is continuous, with no missing data, unlike the GEDI dataset, which has missing data. Another factor is that, when we began the spatialization process, the GEDI database was unavailable for simulation, so we decided to use the Global Forest Canopy database instead. However, if we use the GEDI database in future studies, we expect to increase the carbon sink in the Amazon forest. This is based on our sensitivity analysis and the JULES results in some zones that simulate a high carbon source ($>0.75$ kg C m$^{-2}$ year$^{-1}$) with canopy height $>35$ m, as in the case of Amapá and Acre.

Regarding the question related to the canopy height for the Eddy covariance sites, important to mention that we used this dataset to provide the canopy height in the Table 1.

[Figure]

Figure B: Comparison of GEDI and Global Canopy Forest datasets of Canopy Height and a scatter plot demonstrating the correlation between these two sources in the Brazilian Amazon biome.

In the end of the discussion (section 4.2), we introduced the following sentence

"*Another area for improvement in future studies would be to use other canopy height databases, such as those based on airborne LIDAR observations, to improve the spatialization of plant physiological parameters.* '

- Line 150: Please provide a summary of the basis of European Carbon Tracker estimates, as is done for FluxCom-X.

We would like to thank you for this suggestion. We provided the basis of European Carbon Tracker estimates in the section 2.4

*"European Carbon Tracker provided hourly NEE at a resolution of 0.1° in latitude by 0.2° in longitude, spatial and hourly temporal resolution, calculated by the Simple Biosphere model Version 4 (SiB4), which is driven by meteorology variables from the European Centre for Medium-Range Weather Forecasts (ECMWF) Reanalysis 5th Generation (ERA5) dataset"*

- Line 202: Please describe the scale factor for dark respiration.

We would like to thank you for this suggestion. We added a description of the scale factor for dark respiration in section 2.5.1.

*"Scale factor for dark respiration (fd), which is a coefficient between 0 and 1 associated with leaf dark respiration"*

We also reminded the readers that the equations and a detailed description can be found in the Section 2.1, specifically in the Equation 6 and in the Section 1 of the supplementary materials.

- Lines 293-295: Why was only the area around the ATTO tower used—were data from other areas used for predictions for other areas?

We would like to thank you for this comment. We only mentioned the ATTO site tower; however, the simulations with VPRM were performed for all towers used to calibrate JULES (K67, ATTO, RJA, K34, and the independent tower K83). We added a sentence mentioning the procedure was applied to all towers in section 2.6.

*"These indices are derived from the MODIS Surface Reflectance 8-Day-L3 Global 500 m (MOD09A1) product, which is collected within a ±0.1° area around each tower evaluated in this study (ATTO, K34, K67, K83, RJA; see Table 1 for descriptions)"*

- Line 363-364: The relationship with canopy height and alpha is poor. How important is this?

In fact, the relationship between alpha and canopy height is poor (R² = 0.01). At first glance, this appears to be a limitation, but since alpha varies very little, we are comfortable with this low correlation. Alpha is the third most sensitive parameter in the JULES model, and the value was chosen by varying alpha over a wider range than it actually varies in the Amazon region. To adjust alpha and canopy height, we used the minimum and maximum alpha values presented in the Amazon region, as shown in Figure S3.3.1 of the supplementary material. This is consistent with the range of 0.05 to 0.06 mol mol⁻¹ for C3 species presented by Skilmann (2008). Using alpha with LAI extrapolates this range (see Figure S3.3.1), which includes an increase in value in regions with C4 grasses. However, this is inconsistent with the reduction of this value in plants with this metabolism. Therefore, as alpha is nearly constant, parameterising alpha has a small impact. This could be more important for other vegetation types, which is why we kept it in the adjusted list. We have added the following sentence to Section 3.2 to clarify this aspect:

*"The correlation of alpha with canopy height is small; however, as alpha in the Amazon has a small range of variation (between 0.05 and 0.06 mol mol⁻¹ for C₃ species, in line with Skilman, 2008), this low correlation has a small impact on the final result."*

Technical corrections:

- Line 34: I think this reference should be (Brienen et al., 2015), but I don't see the full citation in the reference section.

Thank you for this correction and we added the correct reference in the line 34 and also in the reference section.

*"The Amazon forest is one of the largest carbon reservoirs in the world, being relevant to the global environment, biodiversity and climate regulation (Brienen et al., 2015)."*

- Line 69: I suggest rewording "having as reference to Eddy-covariance towers in different regions of the Brazilian Amazon biome" to something like "using as references to Eddy-covariance towers in different regions of the Brazilian Amazon biome"

Thank you for this suggestion. We replaced "having" to "using" in the line 70.

*"Here, we present an improvement of the JULES parameterization specifically for the Brazilian Amazon, performing a sensitivity analysis of the model parameters using as reference to Eddy-covariance towers in different regions of the Brazilian Amazon biome"*

- Figure 1: I don't see the black symbol for the validation tower?

In fact, the validation tower was not presented in the Figure 1. We replaced Figure 1 with another version with validation tower K83 in black symbol.

- Line 326: Reword to "The seasonality of the carbon fluxes was not captured by any of the model simulations"

Thank you for the suggestion. We reword the sentence to *"The seasonality of the carbon fluxes was better represented by JULES optimized utilizing the Nelder-Mead method (Figure 3).*

- Line 334: Reword to "direct relation"

Thank you for the suggestion. We replaced the sentence "directly relation" to direct relation". The sentence adapted in the section 3.1:

*"NEE was used as the control variable because it is directly measured in the flux towers "*

**References**

Andreae, M. O., Acevedo, O. C., Araùjo, A., Artaxo, P., Barbosa, C. G. G., Barbosa, H. M. J., Brito, J., Carbone, S., Chi, X., Cintra, B. B. L., da Silva, N. F., Dias, N. L., Dias-Júnior, C. Q., Ditas, F., Ditz, R., Godoi, A. F. L., Godoi, R. H. M., Heimann, M., Hoffmann, T., Kesselmeier, J., Könemann, T., Krüger, M. L., Lavric, J. V., Manzi, A. O., Lopes, A. P., Martins, D. L., Mikhailov, E. F., Moran-Zuloaga, D., Nelson, B. W., Nölscher, A. C., Santos Nogueira, D., Piedade, M. T. F., Pöhlker, C., Pöschl, U., Quesada, C. A., Rizzo, L. V., Ro, C.-U., Ruckteschler, N., Sá, L. D. A., de Oliveira Sá, M., Sales, C. B., dos Santos, R. M. N., Saturno, J., Schöngart, J., Sörgel, M., de Souza, C. M., de Souza, R. A. F., Su, H., Targhetta, N., Tóta, J., Trebs, I., Trumbore, S., van Eijck, A., Walter, D., Wang, Z., Weber, B., Williams, J., Winderlich, J., Wittmann, F., Wolff, S., and Yáñez-Serrano, A. M.: The Amazon Tall Tower Observatory (ATTO): overview of pilot measurements on ecosystem ecology, meteorology, trace gases, and aerosols. Atmospheric Chemistry and Physics, 15(18), 10723-10776. https://doi.org/10.5194/acp-15-10723-2015, 2015.

Brienen, R. J., Phillips, O. L., Feldpausch, T. R., Gloor, E., Baker, T. R., Lloyd, J., .Lopes-Gonzalez, G., Monteagudo-Mendoza, A., Malhi, Y., Lewis, L.S., Vasques Martinez, R., Alexiades, M., Alvarez Davila, E., Alvares Loyaza, P., Andrade, A., Aragão, L.E.O.C., Muramaki, A.A., Arets, E.J.M.M., Arroyo, L., Aymard, C.G.A., Banki, O.S., Baraloto, C., Barroso, J., Donal, D., Zagt, R. J. (2015). Long-term decline of the Amazon carbon sink. *Nature*, *519*(7543), 344-348. https://doi.org/10.1038/nature14283, 2015.

Moraes, R.M., Correa, S. B., Doria, C. D. C., Duponchelle, F., Miranda, G., Montoya, M., ... & ter Steege, H,. Chapter 4: Biodiversity and Ecological Functioning in the Amazon. In:Nobre C, et al (Eds). Amazon Assessment Report 2021. United Nations Sustainable Development Solutions Network, New York, USA. Available from https://www.theamazonwewant.org/spa-reports/. DOI: 10.55161/IKRT9380

Restrepo-Coupe, N., da Rocha, H. R., Hutyra, L. R., de Araujo, A. C., Borma, L. S., Christoffersen, B., and Saleska, S.: LBA-ECO CD-32 Flux Tower Network Data Compilation, Brazilian Amazon: 1999-2006, V2. ORNL DAAC. https://doi.org/10.3334/ORNLDAAC/1842, 2021

Skillman, J. B.: Quantum yield variation across the three pathways of photosynthesis: not yet out of the dark. Journal of experimental botany, 59(7), 1647-1661. https://doi.org/10.1093/jxb/ern029, 2008.

Wallach, D., Makowski, D., Jones, J. W., and Brun, F. (2018). Working with dynamic crop models: methods, tools and examples for agriculture and environment. Academic press.